# Less but More: Linear Adaptive Graph Learning Empowering Spatiotemporal Forecasting

**Jiaming Ma[1], Binwu Wang[1,2,*], Guanjun Wang[1], Kuo Yang[1]**
**Zhengyang Zhou[1,2], Pengkun Wang[1,2], Xu Wang[1,2], Yang Wang[1,2,*]**
[1]University of Science and Technology of China (USTC), Hefei, Anhui, China
[2]Suzhou Institute for Advanced Research, USTC, Suzhou, Jiangsu, China
{JiamingMa, always, yangkuo}@mail.ustc.edu.cn
{wbw2024, zzy0929, pengkun, wx309, angyan}@ustc.edu.cn

## Abstract

The effectiveness of Spatiotemporal Graph Neural Networks (STGNNs) critically hinges on the quality of the underlying graph topology. While end-to-end adaptive graph learning methods have demonstrated promising results in capturing latent spatiotemporal dependencies, they often suffer from high computational complexity and limited expressive capacity. In this paper, we propose MAGE for efficient spatiotemporal forecasting. We first conduct a theoretical analysis demonstrating that the ReLU activation function employed in existing methods amplifies edge-level noise during graph topology learning, thereby compromising the fidelity of the learned graph structures. To enhance model expressiveness, we introduce a sparse yet balanced mixture-of-experts strategy, where each expert perceives the unique underlying graph through kernel-based functions and operates with linear complexity relative to the number of nodes. The sparsity mechanism ensures that each node interacts exclusively with compatible experts, while the balancing mechanism promotes uniform activation across all experts, enabling diverse and adaptive graph representations. Furthermore, we theoretically establish that a single graph convolution using the learned graph in MAGE is mathematically equivalent to multiple convolutional steps under conventional graphs. We evaluate MAGE against advanced baselines on multiple real-world spatiotemporal datasets, and MAGE achieves competitive performance while maintaining strong computational efficiency. Our code is available at official repository.

## 1 Introduction

Spatiotemporal forecasting, a core task in smart city applications, plays a critical role in key domains such as energy, meteorology, and transportation [1, 2, 3]. Among the various approaches, graph-based modeling has become the dominant paradigm for capturing spatiotemporal dependencies, where sensors or base stations are represented as nodes and edges encode spatial and temporal relationships. Spatiotemporal graph neural networks (STGNNs) perform message passing over the graph to learn node representations. As a result, the accuracy of spatiotemporal dependency modeling in these systems critically depends on the quality of the underlying graph structure.

Early graph learning methods typically relied on predefined priors, such as geographic proximity, to compute pairwise node similarities [4, 5, 6]. However, in real-world settings, such prior topological information is often incomplete, noisy, or task-specific [7, 8]. To overcome these limitations, data-driven graph learning methods have emerged as more robust and flexible alternatives, enabling

---

*Binwu Wang and Yang Wang are corresponding authors.

39th Conference on Neural Information Processing Systems (NeurIPS 2025).

end-to-end learning of latent graph structures directly from data. Notable examples include spatial Transformers and adaptive graph learning methods. While offering greater computational efficiency and flexibility than Transformers, the latter has been widely adopted in spatiotemporal forecasting models such as AGCRN [9], GWNet [7], and D$^2$STGNN [10]. These models commonly generate an adjacency matrix $\mathbf{A} = \mathrm{Softmax}(\mathrm{ReLU}(\mathbf{E}_1\mathbf{E}_2^\top))$ through two learnable node embeddings $\mathbf{E}_1$ and $\mathbf{E}_2$. Despite these encouraging results, this method incurs a quadratic complexity with respect to the number of nodes, which limits its scalability on large-scale spatiotemporal systems. Moreover, certain seemingly innocuous but persistently used ReLU activation functions degrade the effectiveness of learning the underlying graph.

To address these limitations, we propose **M**ixture of **A**daptive **G**raph **E**xperts (MAGE), a novel framework that achieves linear computational complexity while offering enhanced expressiveness. First, through extensive theoretical analysis, we reveal that the commonly used ReLU activation function in existing adaptive graph learning method disproportionately amplifies negative edge weights while suppressing positive ones. This behavior inadvertently reinforces noisy edges during graph learning, thereby impairing the model's ability to accurately capture spatiotemporal dependencies—an issue that necessitates removal. Subsequently, we design a kernel-based function as approximation scheme for similarity calculation that reduces the computational complexity from quadratic to linear with respect to the number of nodes. However, according to the matrix theory, we show that such an approximation leads to a reduction in the rank of the learned adjacency matrix, which in turn limits its representational capacity—a so-called low-rank bottleneck. To address this problem, we introduce a sparse yet balanced mixture-of-expert strategy, where each expert learns a distinct adaptive graph. The sparsity strategy enforces that each node interacts only with compatible experts, while the balancing mechanism encourages uniform activation across all experts, thereby facilitating the learning of diverse graph structures. Finally, we provide a theoretical analysis to demonstrate the strong representational capacity of MAGE in capturing adaptive graph structures. MAGE achieves state-of-the-art performance with strong computational efficiency.

Our contributions are summarized as follows,

- ❶ *Practical Solution.* We propose a novel Mixture of Adaptive Graph Experts (MAGE) for efficient spatiotemporal forecasting. MAGE mainly incorporates a sparse yet balanced expert assignment mechanism, where multiple experts interact with nodes in a sparse and selective manner, enabling the extraction of expressive and diverse underlying graph structures.

- ❷ *Theoretical insight.* We provide a comprehensive theoretical foundation for the design motivation in MAGE, such as insights into edge-level noise amplification in existing methods, low-rank bottleneck, and the equivalence between single-step graph convolution in MAGE and multiple convolutions on conventional graphs.

- ❸ *Empirical Study.* Extensive experiments across 17 real-world datasets and 14 advanced baselines show that our method achieves SOTA performance on 94% (48/51) of the metrics while maintaining high computational efficiency and scalability.

## 2 Related Work

spatiotemporal forecasting is a fundamental task in time series analysis and plays a critical role in a wide range of real-world applications [11, 12, 13, 14, 15, 16, 17]. In recent years, STGNNs have become the most representative approach for this task [18, 19, 20]. Early STGNNs relied on static graphs constructed from fixed geographic or domain-specific attributes to capture spatial topology [21, 22]. However, such predefined structures often fail to model the underlying dynamic spatiotemporal dependencies among nodes. To address this limitation, more advanced models have been proposed, such as DGCRN [6], GWNet [7], and D$^2$STGNN [23]. These methods jointly leverage predefined graphs and learnable adaptive graph mechanisms, enabling the model to infer optimal spatial relationships directly from data. Some STGNNs, including AGCRN [9] and MTGNN [24], go even further by completely discarding predefined graphs and relying solely on data-driven adaptive graph structures, thereby achieving strong empirical performance. With the growing popularity of Transformers across various domains, researchers have also developed Transformer-based architectures for spatiotemporal modeling, such as STAEformer [25] and D$^2$STGNN [23]. In this work, we focus on adaptive graph learning, a lightweight yet effective paradigm that captures latent node affinities through a simple dot product between node embeddings. Although this approach is simple

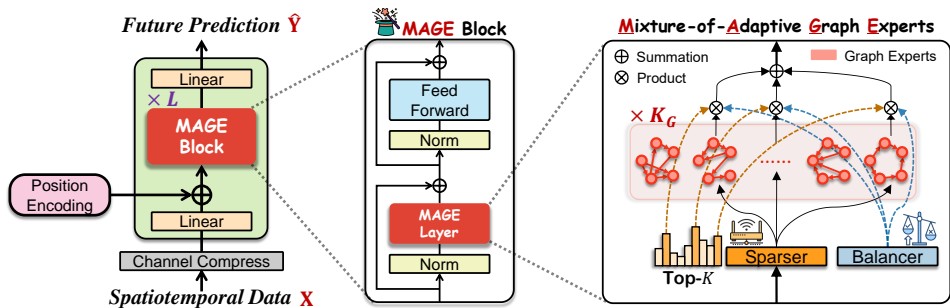

Figure 1: The architecture of MAGE for efficient adaptive graph learning.

and widely adopted in STGNNs, its computational complexity grows quadratically with the number of nodes. To alleviate this scalability bottleneck, BigST [8] introduced positive random features to reduce the graph construction complexity from quadratic to linear in expectation. Building upon BigST, GSNet [26] models the adaptive graph as a low-rank matrix generated via linear transformations. Other methods [27, 28] attempt to prune the learned adaptive graph during inference to reduce computational overhead. Although these approaches improve efficiency by sacrificing some representational capacity, they generally underperform compared to conventional adaptive graph learning techniques.

## 3   Preliminary

**Spatiotemporal Graph.** We use a graph $\mathcal{G} = (\mathcal{V}, \mathcal{E}, \mathbf{A})$ representing spatiotemporal data, where $\mathcal{V}$ means the node set with $N$ nodes, $\mathcal{E}$ represents the set of edges (e.g., similarities between nodes) and $\mathbf{A} \in \mathbb{R}^{N \times N}$ is the weighted adjacency matrix of the graph $\mathcal{G}$ that can be generated based on the static method or data-driven method. We use $x_t \in \mathbb{R}^{N \times f}$ to represent the observed spatiotemporal graph signal at time step $t$, where $f$ indicates the number of feature channels.

**Spatiotemporal forecasting.** Given the graph $\mathcal{G}$ and the historical data of the previous $T$ time steps $\mathbf{X} = \{x_1, \ldots, x_T\} \in \mathbb{R}^{T \times N \times f}$ as input, the goal of spatiotemporal forecasting is to effectively predict future data $\mathbf{Y} = \{x_{T+1}, \ldots, x_{T+T_P}\} \in \mathbb{R}^{T_P \times N \times f}$ in future $T_P$ time steps as output.

**Adaptive Graph Learning.** Adaptive graph learning is typically formulated through a reparameterization of two learnable node embedding matrices, $\mathbf{E}_1, \mathbf{E}_2 \in \mathbb{R}^{N \times d_G}$, where $d_G \ll N$ is the prescribed dimension of the graph generation embeddings. The adaptive graph is then constructed as [7, 9, 23, 24, 29]:

$$\mathbf{A} = \mathrm{Softmax}\left(\mathrm{ReLU}\left(\mathbf{E}_1 \mathbf{E}_2^{\top}\right)\right) \in \mathbb{R}^{N \times N}. \tag{1}$$

Computing the similarity matrix $\mathbf{S} = \mathbf{E}_1 \mathbf{E}_2^{\top} \in \mathbb{R}^{N \times N}$ involves a quadratic time complexity of $\mathcal{O}\left(N^2 d_G\right)$, which significantly hampers the scalability of the method. Moreover, the ReLU function may amplify negative entries in the latent similarity matrix, potentially leading to the unintended enhancement of spurious dependencies (i.e., noise) in the adaptive graph $\mathbf{A}$. We develop an analysis of edge noise amplification theory of this phenomenon, which is provided in Appendix A.1. And we also demonstrate the negative impact of ReLU through extensive experiments in Appendix D.4.

## 4   Methodology

In this section, we present the detailed design of the proposed MAGE. As shown in Figure 1, MAGE first employs a kernel function to reduce the computational complexity of similarity calculation. It then introduces a sparse yet balanced mixture-of-expert method, which adaptively learns the underlying graph topology from the data.

### 4.1   Linear Adaptive Graph Learning

As shown in the theoretical analysis in Appendix A.1, the ReLU function may amplify negative correlations among nodes, potentially leading to the introduction of noise in learned representations.

Thus, we remove $\text{ReLU}\,(\cdot)$ before $\text{Softmax}\,(\cdot)$ to overcome the edge noise issue. that is,

$$\mathbf{A} = \text{Softmax}\left(\mathbf{E}_1 \mathbf{E}_2^\top\right) \in \mathbb{R}^{N \times N}. \tag{2}$$

Thus, the graph convolution of node $v_i$, aggregating from $v_j$, can be defined as,

$$\mathbf{H}_i^{(c)} = \sum_{j \in \mathcal{V}} \frac{\text{Sim}\left(\mathbf{E}_{1i}, \mathbf{E}_{2j}\right) \mathbf{H}_j^{(c-1)}}{\sum_{m \in \mathcal{V}} \text{Sim}\left(\mathbf{E}_{1i}, \mathbf{E}_{2m}\right)} \in \mathbb{R}^d, \tag{3}$$

where $\mathbf{E}_{1i}$ means the $i$-th row of $\mathbf{E}_1$. And $\mathbf{H}_j^{(c-1)}$ means the representation of node $v_j$ in the $(c\text{-}1)$-th graph convolutional layer. $\mathbf{H}_i^{(c)}$ means the output representation of node $v_i$. The above calculation process has quadratic complexity with the number of nodes. The aforementioned computation exhibits quadratic complexity with respect to the number of nodes. To address this limitation, we introduce a kernel-inspired approximation approach [30, 31], which approximates the similarity matrix via the inner product of two non-negative activation functions $\Phi\,(\cdot), \Psi\,(\cdot) : \mathbb{R} \to \mathbb{R}_+ \cup \{0\}$.

$$\mathbf{H}_i^{(c)} = \sum_{j \in \mathcal{V}} \frac{\langle \Phi\left(\mathbf{E}_{1i}\right), \Psi\left(\mathbf{E}_{2j}\right)\rangle \mathbf{H}_j^{(c-1)}}{\sum_{m \in \mathcal{V}} \langle \Phi\left(\mathbf{E}_{1i}\right), \Psi\left(\mathbf{E}_{2m}\right)\rangle} = \frac{\left\langle \Phi\left(\mathbf{E}_{1i}\right), \sum_{j \in \mathcal{V}}\left\langle \Psi\left(\mathbf{E}_{2j}\right), \mathbf{H}_j^{(c-1)}\right\rangle \right\rangle}{\left\langle \Phi\left(\mathbf{E}_{1i}\right), \sum_{m \in \mathcal{V}} \Psi\left(\mathbf{E}_{2m}\right)\right\rangle}. \tag{4}$$

where $\langle \cdot, \cdot \rangle$ is the vector inner product. And we choose exponential activation with bias $\Phi : \mathbf{E}_{1i} \mapsto \exp\left(\mathbf{E}_{1i} + \boldsymbol{\eta}_i\right), \Psi : \mathbf{E}_{2j} \mapsto \exp\left(\mathbf{E}_{2j} + \boldsymbol{\xi}_j\right)$ with all $\boldsymbol{\eta}_i, \boldsymbol{\xi}_j \in \mathbb{R}^{d_G}$ satisfying $\boldsymbol{\eta}_i = \vec{\mathbf{0}}$ and $\xi_{jk} = -\ln\left(\sum_{m \in \mathcal{V}} \exp\left(e_{mk}^{(2)}\right)\right)$. At this point, kernel-based graph convolution can be written as,

$$\mathbf{H}_i^{(c)} = \sum_{j \in \mathcal{V}} \sum_{k=1}^{d_G} \frac{\exp(e_{ik}^{(1)})}{\sum_{w=1}^{d_G} \exp(e_{iw}^{(1)})} \frac{\exp(e_{jk}^{(2)})}{\sum_{m \in \mathcal{V}} \exp(e_{mk}^{(2)})} \mathbf{H}_j^{(c-1)}. \tag{5}$$

And the adaptive graph convolution can be defined as,

$$\mathbf{H}^{(c)} = \text{Softmax}\left(\mathbf{E}_1\right) \text{Softmax}\left(\mathbf{E}_2^\top\right) \mathbf{H}^{(c-1)} \in \mathbb{R}^{N \times d}. \tag{6}$$

where we can first calculate $\text{Softmax}\left(\mathbf{E}_2^\top\right) \mathbf{H}^{(c-1)}$ according to the law of multiplicative union. In this way, the complexity is $\mathcal{O}\left(2 * N * d * d_G\right)$, that is, linearly with the number of nodes $N$, because $d$ and $d_G$ are much smaller than $N$. Detailed derivations of the above part are available in Appendix A.2.

**Low Rank Bottlenecks.** However, the approximate method often incurs degradation in representational capacity. To theoretically characterize this trade-off, we leverage matrix theory, where the rank of the learned adjacency matrix can be used as a measure of the high-dimensional information preserved in the feature representations. Specifically, the rank of the adaptive graph satisfies:

$$\text{Rank}\,(\mathbf{A}) = \text{Rank}\left(\text{Softmax}\left(\mathbf{E}_1\right) \text{Softmax}\left(\mathbf{E}_2^\top\right)\right) \le \min\{N, d_G\} = d_G \ll N. \tag{7}$$

$$\implies \text{Rank}(\mathbf{H}^{(c)}) = \text{Rank}(\mathbf{A}\mathbf{H}^{(c-1)}) \le \min\{d_G, N, d\} = d_G < d. \tag{8}$$

Traditional adaptive graph method can yield graphs with full rank: $\text{Rank}\left(\text{Softmax}\left(\mathbf{E}_1 \mathbf{E}_2^\top\right)\right) = N$, whereas the kernel-based approximate method achieves a lower effective rank. Consequently, the node representations derived via graph convolution are constrained to a low-rank subspace, limiting their expressiveness and discriminative capability.

## 4.2 Mixture-of-Expert for Boosting Linear Adaptive Graph Learning

We introduce the mixture-of-expert strategy, in which each expert independently generates an adaptive graph. At this time, the adaptive graph convolution with $K$ experts can be expressed as:

$$\mathbf{H} = \sum_{k=1}^{K} \alpha_k \mathbf{A}^{(k)} \mathbf{H} = \sum_{k=1}^{K} \alpha_k \text{Softmax}(\mathbf{E}_1^{(k)}) \text{Softmax}(\mathbf{E}_2^{(k)\top}) \mathbf{H} \in \mathbb{R}^{N \times d}. \tag{9}$$

Here $\alpha_k \in [0, 1]$ denotes the sparse yet balanced weights of $K$ experts satisfying $\sum_{k=1}^{K} \alpha_k = 1$. The exact calculation of which will be described in Section 4.2.1. And $\mathbf{E}1^{(k)}$ and $\mathbf{E}2^{(k)}$ denote two

learnable embeddings of the $k$-th expert, which are used to generate its corresponding adaptive graph. At this point, the rank of node representation generated by the mixture-of-expert strategy is expressed as,

$$\text{Rank}(\mathbf{H}^{(c)}) \leq \min\{d, \sum_{k=1}^{K} \min\{d_G, N, d\}\} = \min\{d, Kd_G\} \tag{10}$$

When the number of experts $K$ satisfies $K \geq \lceil d/d_G \rceil$, The rank of the node representation matrix will be bounded by $d$. However, making $K$ too large has little benefit and may even lead to overfitting, as the diversity of features starts to saturate. Therefore, we set $K$ to $\lceil d/d_G \rceil$. At this point, the multi-expert strategy enhances the rank of the representation matrix, which we empirically validate in Section Experiment 5.6.

To further enhance the representational capacity, we incorporate a differential mechanism into the adaptive graph learning process. For each expert $k$, we assign four learnable embeddings: $\mathbf{E}_1^{(k)}, \mathbf{E}_2^{(k)}, \mathbf{E}_3^{(k)}, \mathbf{E}_4^{(k)} \in \mathbb{R}^{N \times d_G}$, and the adaptive graph is generated in a differential manner as follows,

$$\mathbf{A}^{(k)} = \text{Softmax}(\mathbf{E}_1^{(k)})\,\text{Softmax}(\mathbf{E}_2^{(k)\top}) - \lambda\,\text{Softmax}(\mathbf{E}_3^{(k)})\,\text{Softmax}(\mathbf{E}_4^{(k)\top}). \tag{11}$$

To maintain the numerical stability of the $\lambda$, we re-parameterize $\lambda$ as follows,

$$\lambda = \omega + \exp\left(\langle\lambda_1, \lambda_2\rangle\right) - \exp\left(\langle\lambda_3, \lambda_4\rangle\right), \tag{12}$$

where $\omega \in (0, 1)$ is a hyperparameter and $\lambda_1, \lambda_2, \lambda_3, \lambda_4 \in \mathbb{R}^d$ are learnable parameters.

### 4.2.1 Sparse yet Balanced Mixture-of-Expert

We aim to develop a sparse yet balanced mixture-of-experts system. Sparsity ensures that only a small and relevant subset of experts is activated for the input of each node, reducing computational cost. Balance ensures equitable activation across all experts over different inputs, preventing over-reliance on any particular subset. To select the desired $K$ experts, we first define a candidate pool consisting of $K_G > K$ experts, which is denoted as $\mathbb{P} = \{\mathbf{A}^{(k)}\}_{k=1}^{K_G}$.

❶ **Sparse.** For $k$-th expert candidate, we assign learnable identity vectors $\boldsymbol{\theta}_k \in \mathbb{R}^d$, and then we calculate the affinity between the node representation and each expert:

$$\tilde{\alpha}_{ik} = \text{Sigmoid}\left(\mathbf{H}_i^{(c-1)}\boldsymbol{\theta}_k^\top + \gamma_k\right) = \frac{1}{1 + \exp\left(-\gamma_k\right)\exp\left(-\mathbf{H}_i^{(c-1)}\boldsymbol{\theta}_k^\top\right)} = \begin{cases} 1, & \gamma_k \to +\infty, \\ 0, & \gamma_k \to -\infty. \end{cases} \tag{13}$$

where normalized $\tilde{\alpha}_{ik}$ means the affinity between node $i$ and the $k$-th expert candidate. The learnable scalar $\gamma_k \in \mathbb{R}$ with Sigmoid function is used to encourage the model to generate sharply peaked attention weights, favoring clear preferences of candidate experts.

❷ **Balance.** Follow the idea in the work [32], we introduce a priority modulator $\beta$ into the expert selection process described above for balanced activation. If the $k$-th expert candidate is activated more frequently than the average expectation in previous rounds, a negative value $\beta_k$ is applied to penalize its affinity score. Conversely, if it is under-activated, $\beta_k$ takes a positive value to encourage its selection. Accordingly, the optimal expert selection process becomes:

$$\alpha_{ik} = \begin{cases} \tilde{\alpha}_{ik} + \beta_k, & k \in \arg\text{Top-K}\{\tilde{\alpha}_{ir} + \beta_r | r = 1, 2, \ldots, K_G\}, \\ 0, & \text{Otherwise.} \end{cases} \tag{14}$$

where $\arg\text{Top-K}(\cdot)$ means the indices corresponding to the Top-K largest values. i.e., for $K_G$ candidate experts, we retain the Top-K experts that exhibit the highest affinity with node $v_i$. Balanced activations are beneficial for learning generalizable semantic graphs.

Finally, we develop a **load balanced optimization strategy** for $\beta_k$, which computes the difference between the activation count[2] of $k$-th expert $N_k$ and the average activation expectation across $K_G$

---

[2] An expert is considered to be activated once if the attention between it and any node is greater than zero.

candidate experts:

$$\mathcal{L}_{load} = \frac{1}{2} \sum_{k=1}^{K_G} \left| N_k - \frac{(N \cdot K)}{K_G} \right|^2 = \frac{1}{2} \sum_{k=1}^{K_G} \left| \beta_k + \text{StopGrad}\,(N_k - \beta_k) - \frac{(N \cdot K)}{K_G} \right|^2, \quad (15)$$

$$\implies \nabla_{\beta_k} \mathcal{L}_{load} = \frac{1}{2} \nabla_{\beta_k} \left| \beta_k + \text{StopGrad}\,(N_k - \beta_k) - \frac{(N \cdot K)}{K_G} \right|^2 = N_k - \frac{(N \cdot K)}{K_G}. \quad (16)$$

where $\text{StopGrad}(\cdot)$ is the stop-gradient operator [33], keeping the forward output constant but forcing the gradient to zero. Each of the $N$ nodes selects $K$ affinity experts, resulting in a total of $N * K$ expert activations. In this work, we optimize $\beta_k$ by symbolic stochastic gradient descent [34, 35] as follows,

$$\beta_k \leftarrow \beta_k - \mu \,\text{sgn}\,(\nabla_{\beta_k} \mathcal{L}_{load}) = \beta_k - \mu \,\text{sgn}\left( N_k - \frac{(N \cdot K)}{K_G} \right), \quad k = \{1, 2, \ldots, K_G\}. \quad (17)$$

where $\mu > 0$ is the learning rate of optimization of $\beta_k$. $\text{sgn}(\cdot)$ means the signum function. To promote balanced expert utilization, the model adjusts the activation priority of each expert based on its historical usage. Specifically, if the $k$-th expert is selected more frequently than the average, its associated parameter $\beta_k$ is decreased, which reduces its activation probability in subsequent steps.

### 4.3 Mixture of Adaptive Graph Experts

The final version of our adaptive graph convolution of one layer named $\text{MAGE}\,(\cdot)$ is as follows,

$$\text{MAGE}\,(\mathbf{H}) = \sum_{k=1}^{K_G} \text{diag}\,(\alpha_{1k}, \alpha_{2k}, \ldots, \alpha_{Nk})\, \mathbf{A}^{(k)} \mathbf{H}, \quad (18)$$

$$\mathbf{A}^{(k)} = \text{Softmax}(\mathbf{E}_1^{(k)})\,\text{Softmax}(\mathbf{E}_2^{(k)\top}) - \lambda\,\text{Softmax}(\mathbf{E}_3^{(k)})\,\text{Softmax}(\mathbf{E}_4^{(k)\top}). \quad (19)$$

where $\text{diag}\,(\cdot)$ means the diagonal matrix. $\alpha_{ik} \in \mathbb{R}^{N \times K_G}$ denotes the affinity matrix between the node $v_i$ and $k - th$ expert candidate.

### 4.4 Overall Architecture

We stack $L$ layers of MAGE to capture deep-level spatiotemporal dependencies, and the forward process of $l$-th layer can be denoted as follows,

$$\mathbf{Z}^{(l)} = \text{FFN}_l\left( \text{Norm}(\mathbf{H}^{(l)}) \right) + \mathbf{H}^{(l)}, \quad (20)$$

$$\mathbf{H}^{(l)} = \text{MAGE}_l\left( \text{Norm}(\mathbf{Z}^{(l-1)}) \right) + \mathbf{Z}^{(l-1)}, \quad (21)$$

where $\text{FFN}\,(\cdot)$ is the Feed Forward Network with $\text{SwiGLU}\,(\cdot)$ as the activation function [36]. The input representation $\mathbf{Z}^{(0)}$ is the transformation of the input data $\mathbf{X}$ combined with spatiotemporal position embedding as follows,

$$\mathbf{Z}^{(0)} = \mathbf{X}\mathbf{W}_0 + \mathbf{b}_0 + \mathbf{P} \in \mathbb{R}^{N \times d}, \quad (22)$$

where $\mathbf{W}_0$ and $\mathbf{b}_0$ are learnable parameters, and $\mathbf{P} \in \mathbb{R}^{N \times d}$ is the spatiotemporal position embedding, which incorporates various forms of prior information; further details are provided in Appendix B. The final forecasting is generated as follows:

$$\hat{\mathbf{Y}} = \mathbf{Z}^{(L)}\mathbf{W}_{L+1} + \mathbf{b}_{L+1} \in \mathbb{R}^{N \times (T_P * f)}, \quad (23)$$

where $\mathbf{W}_{L+1}$ and $\mathbf{b}_{L+1}$ are learnable parameters. Finally, we redistribute the dimensions of $\hat{\mathbf{Y}}$ to $T_P \times N \times f$ for aligning the dimensions.

## 5 Experiments

### 5.1 Experimental Setup

**Datasets.** We use 18 spatiotemporal datasets from four domains: traffic, energy, meteorology, and mobile communication. Traffic datasets include SD, GBA, GLA and CA in LargeST [37],

XTraffic [38], PeMS series: PeMS0X (X=3,4,7,8) [39] and PeMS-Bay [5] datasets. Energy datasets include Electricity [40] and UrbanEV [41]. Mobile communication datasets include Beijing Weibo, Shanghai Mobile [42], and Milan Internet [43]. Meteorology datasets include KnowAir [44] and China City Air Quality [45]. Details of these datasets are available in Table 4 of Appendix D.1.

**Settings.** Our experiments are deployed on the LargeST platform [37] for all datasets to ensure a fair comparison. All datasets are divided into training, validation, and test sets chronologically in a ratio of 6:2:2. We employ three common metrics: Mean Absolute Error (MAE), Root Mean Square Error (RMSE), and Mean Absolute Percentage Error (MAPE) to evaluate model performance. All experiments are executed on an NVIDIA A100 with 40GB memory. The code environment is based on the PyTorch using Python 3.11.5. The length of the input window and forecasting horizon, $T$ and $T_P$, are set to 12 for all traffic datasets and 24 for other domain datasets. We adopt the Adam [46] optimizer with an $L_1$ loss function, a learning rate of $0.02$, and a predefined milestones decay factor of $0.5$. We use only $L = 3$ MAGE Blocks with hyper residual connections for all experiments, with a dimensionality of $d = 128$ and a graph generation dimension of $d_G = 32$. The maximum number of candidate experts in all datasets $K_G$ is set to 16, and the number of activated experts per node $K$ is set to $\frac{d}{d_G} = 4$. The learning rate for all $\beta_k$ in the load-balanced optimization strategy is $10^{-3}$.

**Baselines.** Our experiments consist of multiple advanced spatiotemporal prediction models, including: AGCRN [9], BigST [8], DGCRN [6], D$^2$STGNN [23], GSNet [26], GWNet [7], MTGNN [24], PatchSTG [47], RPMixer [48], STAEformer [25], STGCN [4], STID [49], STNorm [50], and STWave [51].

## 5.2 Forecasting Performance Comparison

The main results of the forecasting performance comparison are summarized in Table 1. For clarity and readability, we present results on four representative datasets spanning different domains; results on the remaining datasets are provided in Appendix D.2. Methods based on static graph structures, such as STGCN, exhibit limited performance because they cannot capture dynamic spatiotemporal dependencies. GWNet and AGCRN employ adaptive graph learning strategies to improve spatiotemporal modeling. Transformer-based models—D$^2$STGNN, STAEformer, and PatchSTG—are capable of learning adaptive spatiotemporal patterns directly from data, thereby

Table 1: Performance comparisons. The **best** and second best mean performance are in corresponding colors. The '-' marker indicates baseline incur out-of-memory issues even on minimum batch size. The '/' marker indicates baseline is not applicable to this dataset due to the absence of key metadata (e.g., latitude and longitude). All experimental results are the average of five independent runs.

| Method | SD | | | GBA | | | GLA | | | CA | | | XTraffic | | |
|---|---|---|---|---|---|---|---|---|---|---|---|---|---|---|---|
| | MAE | RMSE | MAPE | MAE | RMSE | MAPE | MAE | RMSE | MAPE | MAE | RMSE | MAPE | MAE | RMSE | MAPE |
| STGCN | 19.27 | 33.57 | 13.49 | 23.29 | 38.15 | 17.82 | 22.22 | 37.98 | 14.30 | 20.68 | 35.68 | 15.55 | 13.55 | 26.58 | 31.15 |
| DGCRN | 17.79 | 29.31 | 12.33 | 20.53 | 33.40 | 16.79 | - | - | - | - | - | - | - | - | - |
| AGCRN | 18.39 | 33.63 | 13.78 | 20.69 | 34.30 | 16.05 | 20.26 | 34.86 | 12.39 | - | - | - | - | - | - |
| GWNet | 18.07 | 29.97 | 12.70 | 20.83 | 33.37 | 17.30 | 20.37 | 32.65 | 12.71 | 19.75 | 31.71 | 15.84 | 15.25 | 28.55 | 21.94 |
| MTGNN | 18.21 | 30.99 | 12.36 | 21.48 | 34.91 | 17.17 | 21.75 | 35.35 | 14.88 | 19.91 | 32.63 | 15.11 | 12.48 | 23.39 | 19.50 |
| STNorm | 19.36 | 32.14 | 12.86 | 21.99 | 35.28 | 17.17 | 21.84 | 35.00 | 12.99 | 20.37 | 33.13 | 15.04 | 12.03 | 22.91 | 18.21 |
| STID | 18.03 | 30.85 | 12.18 | 20.65 | 34.29 | 16.92 | 20.40 | 33.90 | 12.97 | 19.04 | 31.86 | 14.69 | 11.62 | 22.41 | 19.84 |
| RPMixer | 26.01 | 43.64 | 18.32 | 28.84 | 52.59 | 26.88 | 28.55 | 51.95 | 19.00 | 25.44 | 47.93 | 20.64 | 16.68 | 43.64 | 32.74 |
| BigST | 17.68 | 29.61 | 11.66 | 21.15 | 34.38 | 17.80 | 20.98 | 34.40 | 13.30 | 19.32 | 32.01 | 14.93 | 12.13 | 23.01 | 21.42 |
| GSNet | 18.75 | 31.30 | 12.67 | 21.88 | 35.38 | 18.04 | 21.31 | 34.75 | 13.46 | 19.60 | 32.24 | 15.30 | 13.35 | 24.87 | 27.09 |
| STWave | 17.64 | 29.61 | 11.83 | 20.56 | 33.58 | 15.14 | 20.22 | 33.03 | 12.38 | 20.67 | 33.12 | 15.76 | - | - | - |
| STAEformer | 19.02 | 31.78 | 12.65 | 21.30 | 34.56 | 17.63 | - | - | - | - | - | - | - | - | - |
| D$^2$STGNN | 17.13 | 28.60 | 12.15 | 21.13 | 34.09 | 16.08 | - | - | - | - | - | - | - | - | - |
| PatchSTG | 17.46 | 30.13 | 11.74 | 19.75 | 33.17 | 14.98 | 19.30 | 32.28 | 11.38 | 17.68 | 29.72 | 12.86 | 10.63 | 20.86 | 19.41 |
| **Ours** | 16.29 | 28.04 | 10.87 | 19.58 | 32.79 | 14.24 | 18.90 | 31.58 | 11.25 | 17.37 | 29.37 | 12.47 | 10.24 | 20.48 | 17.92 |

| Method | Electricity | | | UrbanEV | | | KnowAir | | | China City Air Quality | | | Beijing Weibo | | |
|---|---|---|---|---|---|---|---|---|---|---|---|---|---|---|---|
| | MAE | RMSE | MAPE | MAE | RMSE | MAPE | MAE | RMSE | MAPE | MAE | RMSE | MAPE | MAE | RMSE | MAPE |
| STGCN | 240.2 | 2210 | 14.14 | 5.91 | 12.34 | 19.17 | 15.77 | 24.25 | 57.44 | 19.56 | 33.34 | 28.48 | 0.8549 | 1.6861 | 34.81 |
| DGCRN | 250.3 | 2353 | 18.14 | 5.22 | 11.47 | 18.70 | 21.11 | 30.62 | 65.89 | 21.87 | 35.18 | 35.05 | 0.8637 | 1.7842 | 31.55 |
| AGCRN | 211.5 | 1847 | 16.95 | 5.36 | 12.20 | 18.21 | 16.34 | 24.81 | 63.26 | 19.57 | 32.65 | 31.41 | 0.8505 | 1.6998 | 33.68 |
| GWNet | 200.3 | 1820 | 13.48 | 5.27 | 11.37 | 18.86 | 15.49 | 23.85 | 56.73 | 18.74 | 31.72 | 29.11 | 0.8315 | 1.6777 | 31.74 |
| MTGNN | 194.8 | 1583 | 16.53 | 5.27 | 11.31 | 18.40 | 15.74 | 24.21 | 58.70 | 19.62 | 32.58 | 30.70 | 0.8380 | 1.6653 | 32.59 |
| STNorm | 230.3 | 1983 | 14.92 | 5.43 | 11.54 | 19.24 | 16.00 | 24.32 | 59.46 | 19.72 | 33.13 | 30.04 | 0.8721 | 1.7228 | 32.15 |
| STID | 174.9 | 1532 | 12.48 | 5.23 | 11.39 | 18.24 | 16.16 | 24.88 | 61.41 | 20.54 | 34.13 | 32.86 | 0.8380 | 1.6730 | 32.40 |
| RPMixer | 188.6 | 1574 | 13.19 | 6.52 | 12.62 | 24.80 | 16.73 | 25.96 | 54.07 | 19.05 | 32.46 | 28.91 | 1.0190 | 1.8696 | 45.58 |
| BigST | 190.3 | 1632 | 13.85 | 5.43 | 11.23 | 19.79 | 15.68 | 24.15 | 56.52 | 18.67 | 31.02 | 29.37 | 0.8351 | 1.6806 | 31.32 |
| GSNet | 191.8 | 1617 | 14.98 | 5.55 | 11.39 | 20.26 | 16.30 | 24.68 | 60.37 | 19.50 | 32.04 | 31.29 | 0.8388 | 1.6762 | 32.39 |
| STWave | 188.2 | 1772 | 11.69 | 5.04 | 11.15 | 17.81 | 16.35 | 24.93 | 61.93 | 20.26 | 33.95 | 32.07 | 0.8308 | 1.6849 | 31.28 |
| STAEformer | 200.5 | 1650 | 13.75 | 5.01 | 11.16 | 17.64 | 15.82 | 24.56 | 53.28 | 19.01 | 31.57 | 30.34 | 0.8352 | 1.6810 | 32.12 |
| D$^2$STGNN | 224.8 | 2110 | 17.46 | 5.07 | 11.46 | 17.95 | 15.39 | 24.31 | 55.41 | 18.82 | 32.29 | 26.30 | 0.8489 | 1.7216 | 31.89 |
| PatchSTG | / | / | / | 5.16 | 11.53 | 17.89 | 16.08 | 24.70 | 56.78 | 18.98 | 32.17 | 29.13 | 0.8638 | 1.7561 | 32.16 |
| **Ours** | 172.1 | 1499 | 11.57 | 4.95 | 11.00 | 17.43 | 15.36 | 23.42 | 52.77 | 18.52 | 30.88 | 26.13 | 0.7988 | 1.6477 | 29.85 |

achieving improved performance. However, their high computational complexity hinders scalability, especially on large-scale datasets such as CA and GLA. STID is a linear spatiotemporal modeling architecture that integrates various embedding techniques and achieves performance competitive with GNN-based models. RPMixer captures inter-node relationships through randomly generated projection matrices. DGCRN introduces a dynamic graph that evolves with traffic flow data, but its performance is inconsistent across different scenarios. In contrast, GSNet and BigST adopt enhanced adaptive graph learning mechanisms, achieving both competitive accuracy and good scalability. Our proposed method outperforms all baseline approaches in terms of prediction accuracy, because MAGE enables a more comprehensive exploration of the underlying graph topology, thereby enhancing spatiotemporal modeling and leading to superior forecasting performance.

## 5.3 Ablation Study

In this section, we design following variants of our model to validate the soundness of the main component of our model: '**w/o PE**' removes all the spatiotemporal position encoding embedding; '**w/o SE**' uses only feedforward networks as model backbone without spatial encoder; '**w/o Multi**' leverages only one adaptive graph expert with $K = 1$; '**w/o LB**' reduces the load balanced optimization strategy in MAGE; '**w/o Sparse**' sums up all output of alternative graph convolution. Additionally, the combination ablation experimental results for each spatiotemporal position encoding embeddings are in Figure 4 (b) in Appendix D.3.2. As shown in Figure 2(a), the ablation study reveals that 'w/o SE' variant achieves the worst performance. This is because our mixture-of-adaptive graph convolution module plays a crucial role in guiding the model to recognize dynamics spatiotemporal dependencies among nodes. 'w/o PE' variant also suffers from higher forecasting errors, which can be attributed to the fact that the learnable spatiotemporal position encoding can extract piratical and general knowledge during training. The performance deration of both 'w/o LB' and 'w/o Sparse' variants indicate that sparse and balanced graph convolution possess better performance than dense graph convolution and graph convolution without balancing loading, respectively.

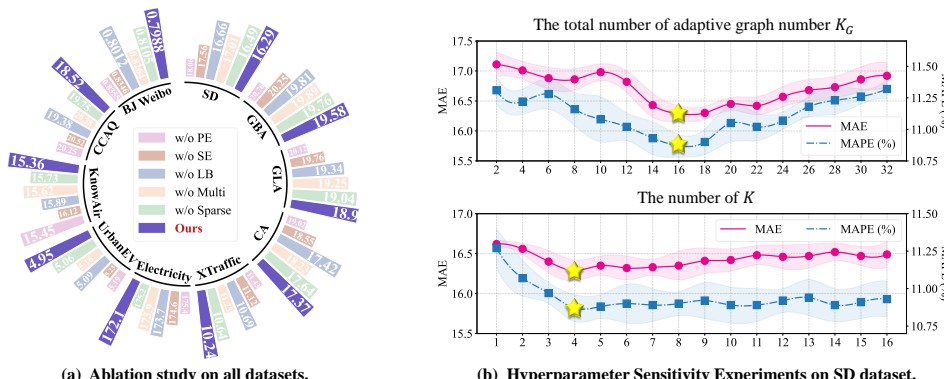

(a) **Ablation study on all datasets.**   (b) **Hyperparameter Sensitivity Experiments on SD dataset.**

Figure 2: (a) Ablation study on all datasets. (b) Hyperparameters sensitivity experiments on $K_G$ and $K$ in SD dataset. The yellow star marks the optimal hyperparameters.

## 5.4 Hyperparameter Sensitivity Experiment

We analyze the sensitivity of MAGE to its two key hyperparameters: the total number of expert graphs $K_G$ and the number of activated experts per node $K$. Using the best-performing configuration as the baseline, we vary one parameter at a time while keeping the others fixed, as shown in Figure 2(b). We report MAE and MAPE on the SD dataset for evaluation. The total number of candidate experts is varied from 2 to 32, and the number of activated experts per node ranges from 1 to 16. When the number of candidate experts $K_G$ is too large, the model struggles to select the most suitable ones, leading to suboptimal performance. Moreover, activating too many experts $K$ each time introduces redundancy and degrades prediction accuracy.

## 5.5 Efficiency Comparison with SOTA STGNNs

As shown in Table 2, our model achieves the highest prediction accuracy among advanced STGNNs while simultaneously demonstrating the lowest computational complexity and the highest efficiency. STWave combines a naive graph convolutional network with decomposition techniques. Although it exhibits relatively high efficiency among the baselines, its performance is only modest. STAEformer employs a standard Transformer architecture with quadratic complexity in the number of nodes, resulting in low efficiency. D$^2$STGNN integrates multiple dynamic graph convolutions, which further increases complexity and leads to training speeds that are more than 118 times slower. On the GBA dataset, it is 960 times slower than our method. In comparison with another advanced model, PatchSTG, MAGE achieves up to a 4.7 times speedup in inference and reduces memory consumption by up to 1.72 times. Moreover, by avoiding complex Transformer architectures that introduce a large number of parameters and high computational costs, our model substantially lowers memory overhead. It requires up to ten times less memory than D$^2$STGNN and STAEformer.

Table 2: Efficiency comparison with SOTA STGNNs. Memory: The maximum memory usage (MB) during training. BS: The maximum allowable batch size during training (up to 64). Train: Average Training Speed (s/epoch). ↑ indicates the relative percentage increasing regarding MAGE.

| Method | SD (716) | | | | GBA (2352) | | | | UrbanEV (1682) | | | |
|---|---|---|---|---|---|---|---|---|---|---|---|---|
| | MAE | Memory | BS | Train | MAE | Memory | BS | Train | MAE | Memory | BS | Train |
| STAEformer | 19.02$_{\uparrow16.75\%}$ | 39,112$_{\uparrow968.05\%}$ | 36$_{\uparrow43.75\%}$ | 384$_{\uparrow1645\%}$ | 21.30$_{\uparrow8.78\%}$ | 39,518$_{\uparrow286.67\%}$ | 5$_{\uparrow92.19\%}$ | 2529$_{\uparrow4336.84\%}$ | 5.09$_{\uparrow1.21\%}$ | 33,680$_{\uparrow502.07\%}$ | 4$_{\uparrow93.75\%}$ | 745$_{\uparrow3625\%}$ |
| STWave | 17.64$_{\uparrow8.28\%}$ | 26,524$_{\uparrow624.30\%}$ | 64$_{\uparrow0.00\%}$ | 411$_{\uparrow1768\%}$ | 20.56$_{\uparrow5.01\%}$ | 40,564$_{\uparrow296.91\%}$ | 26$_{\uparrow59.38\%}$ | 1034$_{\uparrow1714.04\%}$ | 5.04$_{\uparrow1.82\%}$ | 38862$_{\uparrow594.70\%}$ | 18$_{\uparrow71.88\%}$ | 210$_{\uparrow950\%}$ |
| D$^2$STGNN | 17.13$_{\uparrow5.15\%}$ | 40,270$_{\uparrow999.67\%}$ | 31$_{\uparrow51.56\%}$ | 442$_{\uparrow1909\%}$ | 21.13$_{\uparrow7.91\%}$ | 39,102$_{\uparrow282.60\%}$ | 3$_{\uparrow95.31\%}$ | 5527$_{\uparrow9596.49\%}$ | 5.12$_{\uparrow2.42\%}$ | 39006$_{\uparrow597.28\%}$ | 2$_{\uparrow96.875\%}$ | 2257$_{\uparrow11185\%}$ |
| PatchSTG | 17.46$_{\uparrow7.18\%}$ | 7,612$_{\uparrow107.86\%}$ | 64$_{\uparrow0.00\%}$ | 101$_{\uparrow359\%}$ | 19.75$_{\uparrow0.87\%}$ | 27,852$_{\uparrow172.52\%}$ | 64$_{\uparrow0.00\%}$ | 326$_{\uparrow471.93\%}$ | 5.16$_{\uparrow4.24\%}$ | 12,106$_{\uparrow116.41\%}$ | 64$_{\uparrow0.00\%}$ | 25$_{\uparrow25\%}$ |
| **Ours** | **16.29** | **3,662** | **64** | **22** | **19.58** | **10,220** | **64** | **57** | **4.95** | **5594** | **64** | **20** |

## 5.6 Low Rank Bottleneck of Various Models using Adaptive Graph Leaning

In this section, we expose the rank bottleneck in existing adaptive graph learning methods. We compare our model with representative approaches—D$^2$STGNN, BigST, and GSNet-by measuring the effective rank of node representations after graph convolution, computed via SVD with a threshold of $10^{-8}$. To ensure fair comparison across models with varying embedding dimensions, we normalize the rank by its theoretical maximum. As shown in Figure 3 (b), D$^2$STGNN achieves the highest normalized rank (60%), reflecting its strong representational capacity due to standard adaptive graph learning. In contrast, BigST and GSNet adopt linear approximations to reduce computational complexity, resulting in a significant drop in rank (retaining only 20-40% of the theoretical upper bound), which indicates a notable loss of expressive power. Under linear computational complexity, MAGE attains 80% of the theoretical rank limit, outperforming all compared efficient variants. This gain is attributed to its multi-expert adaptive graph mechanism, which supports more diverse and informative spatiotemporal modeling without sacrificing efficiency.

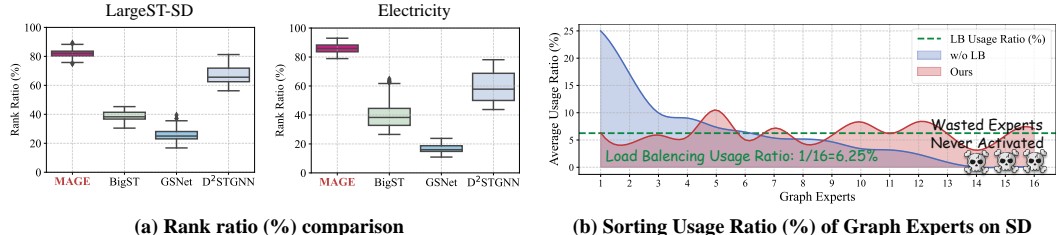

(a) Rank ratio (%) comparison      (b) Sorting Usage Ratio (%) of Graph Experts on SD

Figure 3: (a) Rank ratio (%) comparison on the outputs of adaptive graph convolution which is the ratio of the true rank of the output of the adaptive graph convolution to its rank upper deterministic boundary. (b) Usage Ratio (%) of all graph experts on SD dataset. The expert order is sorting by the usage in 'w/o LB' variant.

## 5.7 Evaluation of the Balancing Strategy in the Mixture-of-Expert System

We evaluate the balancing strategy by analyzing expert utilization on the SD dataset. As shown in Figure 3(b), we compare the full model with a variant that removes the load-balancing component

(w/o LB). For both models, we first extract the affinity scores between nodes and experts, and then count the activation frequency of each of the 16 experts. Our model consists of 16 experts, each activated approximately 6.25% (1/16) of the time, indicating a well-balanced usage across experts. In contrast, the w/o LB variant exhibits highly skewed activation patterns, where certain experts are heavily favored while others are underutilized. This imbalance limits the model's ability to capture diverse spatiotemporal patterns. The balancing mechanism in MAGE ensures a more uniform distribution of expert activations, enabling the learning of richer and more diverse adaptive graph representations, ultimately leading to improved performance.

## 5.8 Pareto-Optimal Trade-off Study Between Linear and Full-Rank Adaptive Graphs

To further examine the efficiency–performance trade-off within MAGE, we conduct a systematic study on the proportion of linear kernel adaptive graphs (Eq. 11) versus naïve full-rank adaptive graphs (Eq. 2) without ReLU. Specifically, we fix the total number of graph experts to 16—consistent with the optimal configuration in the main experiments—and progressively adjust the mixing ratio between the two graph types, while maintaining the original balanced and sparse expert activation constraints.

Table 3: Pareto-optimal study of performance–efficiency trade-offs of adaptive graph type.

| Linear : Full | | MAE | RMSE | MAPE | Memory | Training |
|---|---|---|---|---|---|---|
| **Naïve** | 0:16 | 16.52 | 28.35 | 10.91 | 4,308 MB | 46 s/epoch |
| | 4:12 | 16.53 | 28.29 | 11.01 | 3,998 MB | 35 s/epoch |
| | 8:8 | 17.10 | 28.75 | 11.31 | 3,860 MB | 30 s/epoch |
| | 12:4 | 16.29 | 28.22 | 10.88 | 3,696 MB | 24 s/epoch |
| **Ours** | 16:0 | **16.29** | **28.04** | **10.87** | **3,662** MB | **22** s/epoch |

As shown in Table 3, with the proportion of full-rank adaptive graphs increasing, memory consumption and training time rise substantially, yet without yielding any noticeable performance improvement. In contrast, the pure linear configuration ('Linear:Full' = 16:0, the default setting in MAGE) achieves comparable or even superior forecasting accuracy with minimal resource overhead, indicating that the original MAGE design already lies a Pareto-optimal point in the accuracy and computational efficiency trades-off. Therefore, our linear adaptive graph convolution achieves high predictive performance while maintaining excellent computational efficiency. We further extend the above experiments to investigate the model's inherent preference between linear and full-rank adaptive graph convolutions. The results reveal a clear tendency for the model to favor our proposed linear adaptive graph formulation. Detailed experimental settings, analyses, and results of this study are provided in Appendix D.5.

## 6 Conclusions

In this paper, we propose MAGE, a novel and efficient framework for adaptive graph learning with linear computational complexity. MAGE combines kernel-based approximation with a sparse yet balanced multi-expert architecture. The sparsity mechanism ensures that each node activates only the most relevant experts, while the balancing strategy promotes uniform expert utilization across the network, leading to more robust and representative graph learning. We further provide theoretical insights into the edge noise issue present in existing adaptive graph learning methods. Extensive experiments across multiple spatiotemporal datasets from four distinct domains consistently show that MAGE outperforms state-of-the-art baselines while maintaining excellent computational efficiency.

## Acknowledgment

This paper is partially supported by the National Natural Science Foundation of China (No.12227901). The AI-driven experiments, simulations and model training were performed on the robotic AI-Scientist platform of Chinese Academy of Sciences.

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

# A  Theoretical Justifications

This section presents the theoretical proofs. In the first subsection, we provide a detailed mathematical proof to support our significant observation: *ReLU introduces edge noise in adaptive graph topology generation*. In the second subsection, we mathematically demonstrate the effectiveness of a single convolution on the adaptive adjacency graph convolution, which we refer to as 'You Only Convolve Once' (YOCO). Furthermore, we prove that our adaptive adjacency matrix generation method genuinely achieves YOCO. We pack all the **Theorems** in the red color box, all the **Lemmas** in the cyan color box.

## A.1  Magnify Noisy Edge by ReLU Function in Adaptive Graph Learning Method.

---

**Theorem 1.** *Edge Noise Amplification Theory*

Let $\mathbf{E}_1 = [e_{ik}^{(1)}], \mathbf{E}_2 = [e_{jk}^{(2)}] \in \mathbb{R}^{N \times d_G}$ be the graph generating embeddings where all elements belonging to them satisfy an independent normal distribution $\mathcal{N}\left(0, \sigma^2\right)$ with $\sigma > 0$. $N$ corresponds to the number of nodes and $d_G \ll N$ is the given dimensionality of graph generation embeddings. The Adaptive Graph with or without $\text{ReLU}\left(\cdot\right)$ are respectively calculated as follows,

$$\mathbf{A}^R = \text{Softmax}\left(\text{ReLU}\left(\mathbf{E}_1\mathbf{E}_2^\top\right)\right) \in \mathbb{R}^{N \times N}, \quad \mathbf{A} = \text{Softmax}\left(\mathbf{E}_1\mathbf{E}_2^\top\right) \in \mathbb{R}^{N \times N}. \quad (24)$$

Then, the calculation of Adaptive Graph $\mathbf{A}^R$ will lead to more edge noises than $\mathbf{A}$. Specifically, there exits,
(1) If nodes $i$ and $j$ have positive similarity, then $\mathbf{A}_{ij}^R \leq \mathbf{A}_{ij}$;
(2) If nodes $i$ and $j$ have negative similarity, then with high possibility $\mathbf{A}_{ij}^R \geq \mathbf{A}_{ij}$.

---

*Proof.* (I) We first consider that the nodes $i, j \in \mathcal{V}$ have no negative similarity $s_{ij} = \sum_{k=1}^{d_G} e_{ik}^{(1)} e_{jk}^{(2)} > 0$, and let $\Omega_i = \{l \in \mathcal{V}|s_{il} \geq 0\} \subseteq \mathcal{V}$ be the set of nodes with no negative similarity to node $i$, $\text{ReLU}\left(s_{ij}\right) = \max\left\{s_{ij}, 0\right\}$, then

$$\exp\left(\text{ReLU}\left(s_{ij}\right)\right) = \begin{cases} \exp\left(s_{ij}\right), & j \in \Omega_i, \\ 1, & j \notin \Omega_i, \end{cases} \quad (25)$$

and the final edge weights from node $j$ to node $i$ under two calculation satisfy,

$$\begin{aligned} \mathbf{A}_{ij}^R &= \frac{\exp\left(\text{ReLU}\left(s_{ij}\right)\right)}{\sum_{l \in \Omega_i} \exp\left(\text{ReLU}\left(s_{il}\right)\right)} = \frac{\exp\left(s_{ij}\right)}{\sum_{l \in \Omega_i} \exp\left(s_{il}\right) + \left(N - |\Omega_i|\right)} \\ &\leq \frac{\exp\left(s_{ij}\right)}{\sum_{l \in \Omega_i} \exp\left(s_{il}\right) + \sum_{m \notin \Omega_i} \exp\left(s_{im}\right)} = \frac{\exp\left(s_{ij}\right)}{\sum_{l \in \Omega_i} \exp\left(s_{il}\right)} = \mathbf{A}_{ij}. \end{aligned} \quad (26)$$

If $\exists l^* \in \mathcal{V} - \Omega_i$ such that $s_{il^*} < 0$, i.e., $|\Omega_i| < N$, then the above inequality $\mathbf{A}_{ij}^R < \mathbf{A}_{ij}$ is strictly valid.

(II) Then we consider that the nodes $i, j \in \mathcal{V}$ have negative similarity, i.e. $s_{ij} = \sum_{k=1}^{d_G} e_{ik}^{(1)} e_{jk}^{(2)} < 0$. By Lemma 1 (1), we find $\mathbf{A}_{ij}^R \geq \mathbf{A}_{ij}$ when $s_{ij} \in \left(-\infty, \ln \rho_{ij}\right]$ where $\rho_{ij}$ is the ratio of the sum of the edge weights from nodes other than $j$ to $i$ not containing $\text{ReLU}\left(\cdot\right)$ calculations to the sum of the edge weights from nodes other than $j$ to $i$ containing $\text{ReLU}\left(\cdot\right)$ calculations as follows,

$$\rho_{ij} = \frac{\sum_{l \in \Omega_i} \exp\left(s_{il}\right) + \sum_{m \notin \Omega_i \backslash \{j\}} \exp\left(s_{im}\right)}{\sum_{l \in \Omega_i} \exp\left(s_{il}\right) + \left(N - |\Omega_{ij}| - 1\right)} \in (0, 1]. \quad (27)$$

Although we still find $\mathbf{A}_{ij}^R < \mathbf{A}_{ij}$ when $s_{ij} \in \left(\ln \rho_{ij}, 0\right)$ by Lemma 1 (2), we can make the expectation $\mathbb{E}\left[\rho_{ij}\right]$ asymptotic to 1 for shorting the high possible length of the interval $\left(\ln \rho_{ij}, 0\right)$ approaches 0 by controlling a suitable and reasonable $\sigma$, such as $\sigma = o\left(\sqrt[4]{\left(N - 1\right)/d_G}\right)$ through Lemma 2. Hence, the probability that $s_{ij}$ falls within the interval $\left(\ln \rho_{ij}, 0\right)$ can be controlled to be very low, so that with high probability there is $s_{ij} \in \left(-\infty, \ln \rho_{ij}\right]$ and $\mathbf{A}_{ij}^R \geq \mathbf{A}_{ij}$. $\qquad\square$

**Lemma 1.**

If $i$ and $j$ have negative similarity, then
(1) $\mathbf{A}_{ij}^R \geq \mathbf{A}_{ij}$ when $s_{ij} \in (-\infty, \ln \rho_{ij}]$;
(2) $\mathbf{A}_{ij}^R < \mathbf{A}_{ij}$ when $s_{ij} \in (\ln \rho_{ij}, 0)$,
where $\rho_{ij}$ is the ratio of the sum of the edge weights from nodes other than $j$ to $i$ not containing $\text{ReLU}(\cdot)$ calculations to the sum of the edge weights from nodes other than $j$ to $i$ containing $\text{ReLU}(\cdot)$ calculations as follows,

$$\rho_{ij} = \frac{\sum_{l \in \Omega_i} \exp(s_{il}) + \sum_{m \notin \Omega_i \setminus \{j\}} \exp(s_{im})}{\sum_{l \in \Omega_i} \exp(s_{il}) + (N - |\Omega_{ij}| - 1)} \in (0, 1]. \tag{28}$$

***Proof.*** (I) If $s_{ij} \in (-\infty, \ln \rho_{ij}]$, i.e., $s_{ij} \leq \ln \rho_{ij}$, then

$$\exp(s_{ij}) \leq \rho_{ij} = \frac{\sum_{l \in \Omega_i} \exp(s_{il}) + \sum_{m \notin \Omega_i - \{j\}} \exp(s_{im})}{\sum_{l \in \Omega_i} \exp(s_{il}) + (N - |\Omega_{ij}| - 1)}, \tag{29}$$

$$\Longleftrightarrow \left[\sum_{l \in \Omega_i} \exp(s_{il}) + (N - |\Omega_i| - 1)\right] \exp(s_{ij}) \leq \sum_{l \in \Omega_i} \exp(s_{il}) + \sum_{m \notin \Omega_i \setminus \{j\}} \exp(s_{im}), \tag{30}$$

$$\Longleftrightarrow \left[\sum_{l \in \Omega_i} \exp(s_{il}) + (N - |\Omega_i|)\right] \exp(s_{ij}) \leq \sum_{l \in \Omega_i} \exp(s_{il}) + \sum_{m \notin \Omega_i} \exp(s_{im}), \tag{31}$$

$$\Longleftrightarrow \frac{\exp(s_{ij})}{\sum_{l \in \Omega_i} \exp(s_{il}) + \sum_{m \notin \Omega_i} \exp(s_{im})} \leq \frac{1}{\sum_{l \in \Omega_i} \exp(s_{il}) + (N - |\Omega_i|)}, \tag{32}$$

$$\Longleftrightarrow \mathbf{A}_{ij} \leq \frac{\exp(\text{ReLU}(s_{ij}))}{\sum_{l \in \Omega_i} \exp(\text{ReLU}(s_{il})) + \sum_{m \notin \Omega_i} \exp(\text{ReLU}(s_{im}))}, \tag{33}$$

$$\Longleftrightarrow \mathbf{A}_{ij} \leq \mathbf{A}_{ij}^R. \tag{34}$$

(II) If $s_{ij} \in (\ln \rho_{ij}, 0)$, that is $s_{ij} > \ln \rho_{ij}$, then just reverse the inequality sign from Eq. 29 to 34 can complete the proof. □

**Lemma 2.** *The lower bound of the expectation of negative similarity.*

If $i$ and $j$ have negative similarity, then the expectation $\mathbb{E}[\rho_{ij}]$ satisfies

$$\mathbb{E}[\rho_{ij}] > 1 - \frac{1}{2}\sqrt{\frac{d_G \sigma^4 \left(1 - \frac{1}{4} d_G \sigma^4\right)}{(N-1)}} \to 1^-, \quad \left(\sigma = o(\sqrt[4]{(N-1)/d_G}) \to 0\right). \tag{35}$$

In fact, the above inequality can be further relaxed to

$$\mathbb{E}[\rho_{ij}] > 1 - \frac{1}{2}\sqrt{\frac{d_G \sigma^4 \left(1 - \frac{1}{4} d_G \sigma^4\right)}{(N-1)}} > 1 - \frac{1}{2\sqrt{N-1}} \to 1^{-1}, \quad (N \to +\infty) \tag{36}$$

but this relaxed $\sigma$-independent lower bound is also extremely close to 1 for hundreds to tens of thousands of values of $N$ taken in practice.

***Proof.*** Since all elements in $\mathbf{E}_1, \mathbf{E}_2$ satisfy an independent normal distribution $\mathcal{N}(0, \sigma^2)$, then the expectation of similarity between node $i$ and $j$ is,

$$\mathbb{E}[s_{ij}] = \mathbb{E}\left[\sum_{k=1}^{d_G} e_{ik}^{(1)} e_{jk}^{(2)}\right] = \sum_{k=1}^{d_G} \mathbb{E}\left[e_{ik}^{(1)} e_{jk}^{(2)}\right] = \underbrace{\sum_{k=1}^{d_G} \mathbb{E}\left[e_{ik}^{(1)}\right] \mathbb{E}\left[e_{jk}^{(2)}\right]}_{e_{ik}^{(1)} \text{ and } e_{jk}^{(2)} \text{ are independent for } \forall k=1,2,\dots,d_G.} = \sum_{k=1}^{d_G} 0 = 0, \tag{37}$$

and still since $e_{ik}^{(1)}$ and $e_{jk}^{(2)}$ are independent of each other for arbitrary $k = 1, 2, \ldots, d_G$, the expectation and variance of similarity between node $i$ and $j$ is,

$$
\begin{aligned}
\mathbb{V}\left[s_{ij}\right] =& \mathbb{V}\left[\sum_{k=1}^{d_G} e_{ik}^{(1)} e_{jk}^{(2)}\right] = \sum_{k=1}^{d_G} \mathbb{V}\left[e_{ik}^{(1)} e_{jk}^{(2)}\right] \\
=& \sum_{k=1}^{d_G} \left(\mathbb{V}\left[e_{ik}^{(1)}\right] \mathbb{V}\left[e_{jk}^{(2)}\right] + \mathbb{E}\left[e_{ik}^{(1)}\right]^2 \mathbb{V}\left[e_{jk}^{(2)}\right] + \mathbb{V}\left[e_{ik}^{(1)}\right] \mathbb{E}\left[e_{jk}^{(2)}\right]^2\right) \\
=& \sum_{k=1}^{d_G} \left(\mathbb{V}\left[e_{ik}^{(1)}\right] \mathbb{V}\left[e_{jk}^{(2)}\right] + 0 \cdot \mathbb{V}\left[e_{jk}^{(2)}\right] + \mathbb{V}\left[e_{ik}^{(1)}\right] \cdot 0\right) \\
=& \sum_{k=1}^{d_G} \left(\mathbb{V}\left[e_{ik}^{(1)}\right] \mathbb{V}\left[e_{jk}^{(2)}\right]\right) = \sum_{k=1}^{d_G} \sigma^4 = d_G \sigma^4.
\end{aligned}
\tag{38}
$$

Then we estimate the expectation and variance of edge weights $\exp\left(s_{ij}\right)$ in $\mathrm{Softmax}\left(\cdot\right)$ operation before normalization by an approximation methods based on Taylor series expansions [52] at the expectation $\mathbb{E}\left[s_{ij}\right]$ of $s_{ij}$. Concretely,

$$
\exp\left(s_{ij}\right) = \sum_{r=0}^{+\infty} \frac{\exp\left(\mathbb{E}\left[s_{ij}\right]\right)}{r!} \left(s_{ij} - \mathbb{E}\left[s_{ij}\right]\right)^r = \sum_{r=0}^{+\infty} \frac{\left(s_{ij} - \mathbb{E}\left[s_{ij}\right]\right)^r}{r!},
\tag{39}
$$

$$
\implies \mathbb{E}\left[\exp\left(s_{ij}\right)\right] = \sum_{r=0}^{+\infty} \frac{\mathbb{E}\left[\left(s_{ij} - \mathbb{E}\left[s_{ij}\right]\right)^r\right]}{r!} \approx 1 + \mathbb{E}\left[s_{ij} - \mathbb{E}\left[s_{ij}\right]\right] + \frac{\mathbb{E}\left[\left(s_{ij} - \mathbb{E}\left[s_{ij}\right]\right)^2\right]}{2}
$$

$$
= 1 + 0 + \frac{\mathbb{V}\left[s_{ij}\right]}{2} = 1 + \frac{1}{2} d_G \sigma^4.
\tag{40}
$$

It is important to note that when we take $\sigma = o(d_G^{-\frac{1}{4}})$, $\mathbb{V}\left[s_{ij}\right]$ can be arbitrarily small by choice, so the effect of higher order terms can be ignored and the above approximation will be more accurate.

The equivalence definition of variance is $\mathbb{V}\left[\exp\left(s_{ij}\right)\right] = \mathbb{E}\left[\exp\left(2s_{ij}\right)\right] - \left(\mathbb{E}\left[\exp\left(s_{ij}\right)\right]\right)^2$, hence we use same approximation methods to calculate $\mathbb{E}\left[\exp\left(2s_{ij}\right)\right]$ as follows,

$$
\exp\left(2s_{ij}\right) = \sum_{r=0}^{+\infty} \frac{2^r \exp\left(\mathbb{E}\left[2s_{ij}\right]\right)}{r!} \left(s_{ij} - \mathbb{E}\left[s_{ij}\right]\right)^r = \sum_{r=0}^{+\infty} \frac{2^r \left(s_{ij} - \mathbb{E}\left[s_{ij}\right]\right)^r}{r!},
\tag{41}
$$

$$
\implies \mathbb{E}\left[\exp\left(2s_{ij}\right)\right] = \sum_{r=0}^{+\infty} \frac{2^r \mathbb{E}\left[\left(s_{ij} - \mathbb{E}\left[s_{ij}\right]\right)^r\right]}{r!} \approx 1 + 2\mathbb{E}\left[s_{ij} - \mathbb{E}\left[s_{ij}\right]\right] + 2\mathbb{E}\left[\left(s_{ij} - \mathbb{E}\left[s_{ij}\right]\right)^2\right]
$$

$$
= 1 + 0 + 2\mathbb{V}\left[s_{ij}\right] = 1 + 2d_G \sigma^4.
\tag{42}
$$

Then the variance of $\exp\left(s_{ij}\right)$ is,

$$
\mathbb{V}\left[\exp\left(s_{ij}\right)\right] = \mathbb{E}\left[\exp\left(2s_{ij}\right)\right] - \left(\mathbb{E}\left[\exp\left(s_{ij}\right)\right]\right)^2 \approx \left(1 + 2d_G \sigma^4\right) - \left(1 + \frac{1}{2} d_G \sigma^4\right)^2
\tag{43}
$$

$$
= d_G \sigma^4 - \frac{1}{4} d_G^2 \sigma^8 = d_G \sigma^4 \left(1 - \frac{1}{4} d_G \sigma^4\right).
\tag{44}
$$

Let $\zeta_{ij} = \sum_{l \in \Omega_i} \exp\left(s_{il}\right) + \sum_{m \notin \Omega_i \setminus \{j\}} \exp\left(s_{im}\right)$ and $\zeta_{ij}^R = \sum_{l \in \Omega_i} \exp\left(s_{il}\right) + \left(N - |\Omega_i| - 1\right)$, then the definition of $\rho_{ij}$ is,

$$
\rho_{ij} = \frac{\zeta_{ij}}{\zeta_{ij}^R} \in (0, 1].
\tag{45}
$$

Then we can get that,

$$\mathbb{E}\left[\zeta_{ij}\right] = \sum_{l \neq j} \mathbb{E}\left[\exp\left(s_{il}\right)\right] = (N-1)\left(1 + \frac{1}{2}d_G\sigma^4\right), \tag{46}$$

$$\mathbb{V}\left[\zeta_{ij}\right] = \sum_{l \neq j} \mathbb{V}\left[\exp\left(s_{il}\right)\right] = (N-1)d_G\sigma^4\left(1 - \frac{1}{4}d_G\sigma^4\right), \tag{47}$$

$$\mathbb{E}\left[\zeta_{ij}^R\right] = \sum_{l \in \Omega_i} \mathbb{E}\left[\exp\left(s_{il}\right)\right] + (N - |\Omega_i| - 1) = |\Omega_i|\left(1 + \frac{1}{2}d_G\sigma^4\right) + (N - |\Omega_i| - 1) \tag{48}$$

$$\mathbb{V}\left[\zeta_{ij}^R\right] = \sum_{l \in \Omega_i} \mathbb{V}\left[\exp\left(s_{il}\right)\right] = |\Omega_i|d_G\sigma^4\left(1 - \frac{1}{4}d_G\sigma^4\right). \tag{49}$$

The expectation of $\rho_{ij}$ can be expressed as,

$$\begin{aligned}
\mathbb{E}\left[\rho_{ij}\right] &= \mathbb{E}\left[\zeta_{ij}/\zeta_{ij}^R\right] = \mathbb{E}\left[\zeta_{ij} \cdot 1/\zeta_{ij}^R\right] = \mathbb{E}\left[\zeta_{ij}\right]\mathbb{E}\left[1/\zeta_{ij}^R\right] + \mathrm{Cov}\left(\zeta_{ij}, 1/\zeta_{ij}^R\right) \\
&= \mathbb{E}\left[\zeta_{ij}\right]\mathbb{E}\left[1/\zeta_{ij}^R\right] + \mathrm{Corr}\left(\zeta_{ij}, 1/\zeta_{ij}^R\right)\sqrt{\mathbb{V}\left[\zeta_{ij}\right]\mathbb{V}\left[1/\zeta_{ij}^R\right]} \\
&\geq \mathbb{E}\left[\zeta_{ij}\right]\mathbb{E}\left[1/\zeta_{ij}^R\right] - \sqrt{\mathbb{V}\left[\zeta_{ij}\right]\mathbb{V}\left[1/\zeta_{ij}^R\right]}.
\end{aligned} \tag{50}$$

Where $\mathrm{Cov}\left(\zeta_{ij}, 1/\zeta_{ij}^R\right)$ is the covariance between $\zeta_{ij}$ and $1/\zeta_{ij}^R$, and $\mathrm{Corr}\left(\zeta_{ij}, 1/\zeta_{ij}^R\right) \in [-1, 1]$ is the correlation coefficient between $\zeta_{ij}$ and $1/\zeta_{ij}^R$. We compute $\mathbb{E}\left[1/\zeta_{ij}^R\right]$ and $\mathbb{V}\left[1/\zeta_{ij}^R\right]$ by a similar Taylor expansion approximation as above,

$$1/\zeta_{ij}^R = \sum_{r=0}^{+\infty} \frac{(-1)^r}{\mathbb{E}\left[\zeta_{ij}^R\right]^{r+1}}\left(\zeta_{ij}^R - \mathbb{E}\left[\zeta_{ij}^R\right]\right)^r, \tag{51}$$

$$\begin{aligned}
\implies \mathbb{E}\left[1/\zeta_{ij}^R\right] &= \sum_{r=0}^{+\infty} \frac{(-1)^r}{\mathbb{E}\left[\zeta_{ij}^R\right]^{r+1}}\mathbb{E}\left[\left(\zeta_{ij}^R - \mathbb{E}\left[\zeta_{ij}^R\right]\right)^r\right] \\
&\approx \frac{1}{\mathbb{E}\left[\zeta_{ij}^R\right]} - \frac{\mathbb{E}\left[\left(\zeta_{ij}^R - \mathbb{E}\left[\zeta_{ij}^R\right]\right)\right]}{\mathbb{E}\left[\zeta_{ij}^R\right]^2} + \frac{\mathbb{E}\left[\left(\zeta_{ij}^R - \mathbb{E}\left[\zeta_{ij}^R\right]\right)^2\right]}{\mathbb{E}\left[\zeta_{ij}^R\right]^3} \\
&= \frac{1}{\mathbb{E}\left[\zeta_{ij}^R\right]} - 0 + \frac{\mathbb{V}\left[\zeta_{ij}^R\right]}{\mathbb{E}\left[\zeta_{ij}^R\right]^3} \\
&= \frac{1}{\mathbb{E}\left[\zeta_{ij}^R\right]}\left(1 + \frac{\mathbb{V}\left[\zeta_{ij}^R\right]}{\mathbb{E}\left[\zeta_{ij}^R\right]^2}\right) \\
&= \frac{1}{\mathbb{E}\left[\zeta_{ij}^R\right]}\left(1 + \tau_{ij}\right).
\end{aligned} \tag{52}$$

where $\tau_{ij} = \mathbb{V}\left[\zeta_{ij}^R\right]\mathbb{E}\left[\zeta_{ij}^R\right]^{-2}$. Similarity there is $\mathbb{V}\left[1/\zeta_{ij}^R\right] = \mathbb{E}\left[\left(1/\zeta_{ij}^R\right)^2\right] - \left(\mathbb{E}\left[1/\zeta_{ij}^R\right]\right)^2$, and the $\mathbb{E}\left[\left(1/\zeta_{ij}^R\right)^2\right]$ can be calculated as,

$$\left(1/\zeta_{ij}^R\right)^2 = \sum_{r=0}^{+\infty} \frac{(-1)^r (r+1)}{\mathbb{E}\left[\zeta_{ij}^R\right]^{r+2}} \left(\zeta_{ij}^R - \mathbb{E}\left[\zeta_{ij}^R\right]\right)^r, \tag{53}$$

$$\implies \quad \mathbb{E}\left[\left(1/\zeta_{ij}^R\right)^2\right] = \sum_{r=0}^{+\infty} \frac{(-1)^r (r+1)}{\mathbb{E}\left[\zeta_{ij}^R\right]^{r+2}} \mathbb{E}\left[\left(\zeta_{ij}^R - \mathbb{E}\left[\zeta_{ij}^R\right]\right)^r\right]$$

$$\approx \frac{1}{\mathbb{E}\left[\zeta_{ij}^R\right]^2} - \frac{2\mathbb{E}\left[\left(\zeta_{ij}^R - \mathbb{E}\left[\zeta_{ij}^R\right]\right)\right]}{\mathbb{E}\left[\zeta_{ij}^R\right]^3} + \frac{3\mathbb{E}\left[\left(\zeta_{ij}^R - \mathbb{E}\left[\zeta_{ij}^R\right]\right)^2\right]}{\mathbb{E}\left[\zeta_{ij}^R\right]^4}$$

$$= \frac{1}{\mathbb{E}\left[\zeta_{ij}^R\right]^2} - 0 + \frac{3\mathbb{V}\left[\zeta_{ij}^R\right]}{\mathbb{E}\left[\zeta_{ij}^R\right]^4}$$

$$= \frac{1}{\mathbb{E}\left[\zeta_{ij}^R\right]^2} \left(1 + 3\frac{\mathbb{V}\left[\zeta_{ij}^R\right]}{\mathbb{E}\left[\zeta_{ij}^R\right]^2}\right) \tag{54}$$

$$= \frac{1}{\mathbb{E}\left[\zeta_{ij}^R\right]^2} \left(1 + 3\tau_{ij}\right). \tag{55}$$

Then the variance of $1/\zeta_{ij}^R$ can be expressed as,

$$\mathbb{V}\left[1/\zeta_{ij}^R\right] = \mathbb{E}\left[\left(1/\zeta_{ij}^R\right)^2\right] - \left(\mathbb{E}\left[1/\zeta_{ij}^R\right]\right)^2 = \frac{(1+3\tau_{ij}) - (1+\tau_{ij})^2}{\mathbb{E}\left[\zeta_{ij}^R\right]^2} = \frac{\tau_{ij}(1-\tau_{ij})}{\mathbb{E}\left[\zeta_{ij}^R\right]^2} \tag{56}$$

**Claim.1 in Lemma 2:** $\mathbb{E}\left[\zeta_{ij}\right]\mathbb{E}\left[1/\zeta_{ij}^R\right] > 1$. In fact,

$$\mathbb{E}\left[\zeta_{ij}\right]\mathbb{E}\left[1/\zeta_{ij}^R\right] = (N-1)\left(1 + \frac{1}{2}d_G\sigma^4\right)\frac{1}{\mathbb{E}\left[\zeta_{ij}^R\right]}\left(1 + \frac{\mathbb{V}\left[\zeta_{ij}^R\right]}{\mathbb{E}\left[\zeta_{ij}^R\right]^2}\right)$$

$$\geq (N-1)\left(1 + \frac{1}{2}d_G\sigma^4\right)\frac{1}{\mathbb{E}\left[\zeta_{ij}^R\right]} \cdot 1$$

$$= \frac{(N-1)\left(1 + \frac{1}{2}d_G\sigma^4\right)}{|\Omega_i|\left(1 + \frac{1}{2}d_G\sigma^4\right) + (N - |\Omega_i| - 1)}$$

$$= \frac{(N-1)}{|\Omega_i| + \left(\frac{N - |\Omega_i| - 1}{1 + \frac{1}{2}d_G\sigma^4}\right)} \tag{57}$$

$$> \frac{(N-1)}{|\Omega_i| + \left(\frac{N - |\Omega_i| - 1}{1}\right)}$$

$$= \frac{(N-1)}{(N-1)} = 1.$$

The first inequality in above Eq. 57 holds since $\mathbb{V}\left[\zeta_{ij}^R\right] = |\Omega_i|\left(1 + 2d_G\sigma^4\right) > 0$ and $\mathbb{E}\left[\zeta_{ij}^R\right]^2 \geq 0$.

**Claim.2 in Lemma 2:** $\mathbb{V}\left[\zeta_{ij}\right]\mathbb{V}\left[1/\zeta_{ij}^{R}\right] < \frac{d_{G}\sigma^{4}\left(1-\frac{1}{4}d_{G}\sigma^{4}\right)}{4(N-1)}$. In fact,

$$
\begin{aligned}
\mathbb{V}\left[\zeta_{ij}\right]\mathbb{V}\left[1/\zeta_{ij}^{R}\right] &= (N-1)\,d_{G}\sigma^{4}\left(1-\frac{1}{4}d_{G}\sigma^{4}\right)\frac{\tau_{ij}\left(1-\tau_{ij}\right)}{\mathbb{E}\left[\zeta_{ij}^{R}\right]^{2}}\\
&= \frac{(N-1)\,d_{G}\sigma^{4}\left(1-\frac{1}{4}d_{G}\sigma^{4}\right)\tau_{ij}\left(1-\tau_{ij}\right)}{\left[|\Omega_{i}|\left(1+\frac{1}{2}d_{G}\sigma^{4}\right)+(N-|\Omega_{i}|-1)\right]^{2}}\\
&= \frac{(N-1)\,d_{G}\sigma^{4}\left(1-\frac{1}{4}d_{G}\sigma^{4}\right)\tau_{ij}\left(1-\tau_{ij}\right)}{\left[|\Omega_{i}|\left(\frac{1}{2}d_{G}\sigma^{4}\right)+(N-1)\right]^{2}}\\
&< \frac{(N-1)\,d_{G}\sigma^{4}\left(1-\frac{1}{4}d_{G}\sigma^{4}\right)}{4\,(N-1)^{2}}\\
&= \frac{d_{G}\sigma^{4}\left(1-\frac{1}{4}d_{G}\sigma^{4}\right)}{4\,(N-1)}\\
&\leq \frac{1}{4\,(N-1)}.
\end{aligned}
\tag{58}
$$

The first inequality holds since the $|\Omega_{i}|$ in the denominator is no less than 0, and the maximum tight value-independent upper bound of formula $\tau_{ij}\left(1-\tau_{ij}\right)$ is $1/4$. The second inequality holds since the maximum tight value-independent upper bound of formula $d_{G}\sigma^{4}\left(1-\frac{1}{4}d_{G}\sigma^{4}\right)$ is 1.

By above **Claim.1** and **2**, we have $\mathbb{E}\left[\zeta_{ij}\right]\mathbb{E}\left[1/\zeta_{ij}^{R}\right] > 1$ and $\mathbb{V}\left[\zeta_{ij}\right]\mathbb{V}\left[1/\zeta_{ij}^{R}\right] < \frac{d_{G}\sigma^{4}\left(1-\frac{1}{4}d_{G}\sigma^{4}\right)}{4(N-1)}$. Finally we obtain

$$
\mathbb{E}\left[\rho_{ij}\right] > 1-\frac{1}{2}\sqrt{\frac{d_{G}\sigma^{4}\left(1-\frac{1}{4}d_{G}\sigma^{4}\right)}{(N-1)}}.
\tag{59}
$$

$\square$

## A.2 Scalability Graph Reparametrization Generation Enpowered by Kernel-like Method

We focus only on operations with the convolution procedure on adaptive graph, i.e.,

$$
\mathbf{A}\star_{\mathcal{G}}\mathbf{H}^{(l)} = \mathrm{Softmax}\left(\mathbf{E}_{1}\mathbf{E}_{2}^{\top}\right)\mathbf{H}^{(l)}.
\tag{60}
$$

We cancel the calculation $\mathrm{ReLU}\left(\cdot\right)$ before $\mathrm{Softmax}\left(\cdot\right)$ to overcome the edge noises issue, that is,

$$
\mathbf{A} = \mathrm{Softmax}\left(\mathbf{E}_{1}\mathbf{E}_{2}^{\top}\right) \in \mathbb{R}^{N\times N}
\tag{61}
$$

The calculation complexity of Eq. 61 is still $\mathcal{O}\left(d_{G}N^{2}\right)$ since the non-linear operation $\mathrm{Softmax}\left(\cdot\right)$ forbids the law of union for multiplication among three matrices $\mathbf{E}_{1}$, $\mathbf{E}_{2}^{\top}$ and $\mathbf{H}^{(l)}$, and the similarity matrix $\mathbf{S} = \mathbf{E}_{1}\mathbf{E}_{2}^{\top} \in \mathbb{R}^{N\times N}$ need to be counted and cost $\mathcal{O}\left(d_{G}N^{2}\right)$ complexity. Then we introduce a kind of kernel-like method to linearizable simplification for adaptive matrix. A one layer adaptive graph convolution without parameters can be expressed as follows,

$$
\mathbf{H}^{(l+1)} = \mathbf{A}\star_{\mathcal{G}}\mathbf{H}^{(l)} = \mathbf{A}\mathbf{H}^{(l)}
\tag{62}
$$

We consider the representation of node $i \in \mathcal{V}$ after adaptive graph convolution, i.e., the $i$-th row of $\mathbf{A}\star_{\mathcal{G}}\mathbf{H}^{(l)}$ as follows,

$$
\left(\mathbf{A}\mathbf{H}^{(l)}\right)[i] = \sum_{j\in\mathcal{V}}\frac{\exp\left(\sum_{k=1}^{d_{G}}e_{ik}^{(1)}e_{jk}^{(2)}\right)\mathbf{H}_{j}^{(l)}}{\sum_{m\in\mathcal{V}}\exp\left(\sum_{k=1}^{d_{G}}e_{ik}^{(1)}e_{mk}^{(2)}\right)} \in \mathbb{R}^{d}
\tag{63}
$$

Hence the adaptive graph convolution is actually equivalent to a $l_{1}$ weighted average of the spatiotemporal representation of nodes with weights $\exp\left(\sum_{k=1}^{d_{G}}e_{ik}^{(1)}e_{jk}^{(2)}\right) = \exp\left(\langle\mathbf{e}_{i}^{(1)},\mathbf{e}_{j}^{(2)}\rangle\right) > 0$ where $\langle\cdot,\cdot\rangle$ is the vector inner product, which is the exponential activation of the inner product of two graph

generating embeddings $\mathbf{E}_1, \mathbf{E}_2$ corresponding to different nodes. We can define an universe form of adaptive graph convolution as follows,

$$\left(\mathbf{A}\mathbf{H}^{(l)}\right)[i] = \sum_{j \in \mathcal{V}} \frac{\mathrm{Sim}(\mathbf{e}_i^{(1)}, \mathbf{e}_j^{(2)})\mathbf{H}_j^{(l)}}{\sum_{m \in \mathcal{V}} \mathrm{Sim}(\mathbf{e}_i^{(1)}, \mathbf{e}_m^{(2)})} \in \mathbb{R}^d, \tag{64}$$

where the binary function $\mathrm{Sim}(\cdot, \cdot) : \mathbb{R}^{d_G} \times \mathbb{R}^{d_G} \to \mathbb{R}_+ \cup \{0\}$ is a positive-definite kernel, computing the similarity between two embeddings as the weights. Take the example in Eq. 63 above, i.e., $\mathrm{Sim}(\mathbf{e}_i^{(1)}, \mathbf{e}_j^{(2)}) = \exp(\langle \mathbf{e}_i^{(1)}, \mathbf{e}_j^{(2)} \rangle)$, the exponential activation after inner product computation guarantees the non-negativity of similarity, but it also restricts the implementation of the multiplicative union law, leading to the introduction of high complexity. We therefore draw on the kernel-like method [30, 31] to ensure non-negativity of similarity by introducing a non-negative activation function before the inner product computation. The objective is to eliminate the activation operation following the inner product calculation, whilst preserving the non-negativity of the similarity calculation. When all elements of the object of inner product satisfy non-negativity, the result of the inner product is naturally non-negative. To this end, two non-negative activation functions $\Phi(\cdot), \Psi(\cdot) : \mathbb{R} \to \mathbb{R}_+ \cup \{0\}$ as kernel-functions are incorporated prior to the inner product, thereby ensuring that all elements of the inner product object satisfy non-negativity.

$$\mathrm{Sim}(\mathbf{e}_i^{(1)}, \mathbf{e}_j^{(2)}) = \left\langle \Phi(\mathbf{e}_i^{(1)}), \Psi(\mathbf{e}_j^{(2)}) \right\rangle \tag{65}$$

Thus by using the above kernel-like method we can rewrite the adaptive graph convolution in the form as follows:

$$\left(\mathbf{A} \star_{\mathcal{G}} \mathbf{H}^{(l)}\right)[i] = \sum_{j \in \mathcal{V}} \frac{\left\langle \Phi(\mathbf{e}_i^{(1)}), \Psi(\mathbf{e}_j^{(2)}) \right\rangle \mathbf{H}_j^{(l)}}{\sum_{m \in \mathcal{V}} \left\langle \Phi(\mathbf{e}_i^{(1)}), \Psi(\mathbf{e}_m^{(2)}) \right\rangle} = \sum_{j \in \mathcal{V}} \frac{\left\langle \Phi(\mathbf{e}_i^{(1)}), \langle \Psi(\mathbf{e}_j^{(2)}), \mathbf{H}_j^{(l)} \rangle \right\rangle}{\left\langle \Phi(\mathbf{e}_i^{(1)}), \sum_{m \in \mathcal{V}} \Psi(\mathbf{e}_m^{(2)}) \right\rangle}$$

$$= \frac{\left\langle \Phi(\mathbf{e}_i^{(1)}), \sum_{j \in \mathcal{V}} \langle \Psi(\mathbf{e}_j^{(2)}), \mathbf{H}_j^{(l)} \rangle \right\rangle}{\left\langle \Phi(\mathbf{e}_i^{(1)}), \sum_{m \in \mathcal{V}} \Psi(\mathbf{e}_m^{(2)}) \right\rangle} \in \mathbb{R}^d, \tag{66}$$

We are therefore able to use the law of multiplicative union to prioritize the inner product of $\Psi(\mathbf{e}_j^{(2)})$ and $\mathbf{H}_j^{(l)}$. Here we choose differentiable weighted nonlinear functions $\Phi : \mathbf{e}_i^{(1)} \mapsto \exp(\mathbf{e}_i^{(1)} + \boldsymbol{\eta}_i), \Psi : \mathbf{e}_j^{(2)} \mapsto \exp(\mathbf{e}_j^{(2)} + \boldsymbol{\xi}_j)$ with all $\boldsymbol{\eta}_i, \boldsymbol{\xi}_j \in \mathbb{R}^{d_G}$, then

We are therefore able to use the law of multiplicative union to prioritize the inner product of $\Psi(\mathbf{e}_j^{(2)})$ and $\mathbf{H}_j^{(l)}$. Here we choose differentiable weighted nonlinear functions $\Phi : \mathbf{e}_i^{(1)} \mapsto \exp(\mathbf{e}_i^{(1)} + \boldsymbol{\eta}_i), \Psi : \mathbf{e}_j^{(2)} \mapsto \exp(\mathbf{e}_j^{(2)} + \boldsymbol{\xi}_j)$ with all $\boldsymbol{\eta}_i, \boldsymbol{\xi}_j \in \mathbb{R}^{d_G}$, then

$$\left(\mathbf{A} \star_{\mathcal{G}} \mathbf{H}^{(l)}\right)[i] = \frac{\left\langle \exp\left(\mathbf{e}_i^{(1)} + \boldsymbol{\eta}_i\right), \sum_{j \in \mathcal{V}} \left\langle \exp\left(\mathbf{e}_j^{(2)} + \boldsymbol{\xi}_j\right), \mathbf{H}_j^{(l)} \right\rangle \right\rangle}{\left\langle \exp\left(\mathbf{e}_i^{(1)} + \boldsymbol{\eta}_i\right), \sum_{m \in \mathcal{V}} \exp\left(\mathbf{e}_m^{(2)} + \boldsymbol{\xi}_m\right) \right\rangle}$$

$$= \frac{\sum_{k=1}^{d_G} \exp\left(e_{ik}^{(1)} + \eta_{ik}\right)\left(\sum_{j \in \mathcal{V}} \left\langle \exp\left(\mathbf{e}_j^{(2)} + \boldsymbol{\xi}_j\right), \mathbf{H}_j^{(l)} \right\rangle\right)[k]}{\sum_{w=1}^{d_G} \exp\left(e_{iw}^{(1)} + \eta_{iw}\right) \sum_{m \in \mathcal{V}} \exp\left(e_{mw}^{(2)} + \xi_{mw}\right)}$$

$$= \frac{\sum_{k=1}^{d_G} \exp\left(e_{ik}^{(1)} + \eta_{ik}\right) \sum_{j \in \mathcal{V}} \exp\left(e_{jk}^{(2)} + \xi_{jk}\right) \mathbf{H}_j^{(l)}}{\sum_{m \in \mathcal{V}} \sum_{w=1}^{d_G} \exp\left(e_{iw}^{(1)} + e_{mw}^{(2)} + \eta_{iw} + \xi_{mw}\right)} \tag{67}$$

$$= \sum_{j \in \mathcal{V}} \frac{\sum_{k=1}^{d_G} \exp\left(e_{ik}^{(1)} + e_{jk}^{(2)} + \eta_{ik} + \xi_{jk}\right) \mathbf{H}_j^{(l)}}{\sum_{m \in \mathcal{V}} \sum_{w=1}^{d_G} \exp\left(e_{iw}^{(1)} + e_{mw}^{(2)} + \eta_{iw} + \xi_{mw}\right)}.$$

Hence this kind of adaptive graph convolution is actually equivalent to a $l_1$ weighted average of the spatiotemporal representation of nodes with weights $\sum_{k=1}^{d_G} \exp\left(e_{ik}^{(1)} + e_{jk}^{(2)} + \eta_{ik} + \xi_{jk}\right) > 0$.

Once we know how the elements are computed in Eq. 67, we wish to rewrite the equation into the general matrix computation like Eq. 60. as follows,

$$
\begin{aligned}
\left(\mathbf{A} \star_{\mathcal{G}} \mathbf{H}^{(l)}\right)[i] &= \sum_{j \in \mathcal{V}} \frac{\sum_{k=1}^{d_G} \exp\left(e_{ik}^{(1)} + e_{jk}^{(2)} + \eta_{ik} + \xi_{jk}\right) \mathbf{H}_j^{(l)}}{\sum_{m \in \mathcal{V}} \sum_{w=1}^{d_G} \exp\left(e_{iw}^{(1)} + e_{mw}^{(2)} + \eta_{iw} + \xi_{mw}\right)} \\
&= \sum_{j \in \mathcal{V}} \sum_{k=1}^{d_G} \frac{\exp\left(e_{ik}^{(1)} + \eta_{ik}\right) \exp\left(e_{jk}^{(2)} + \xi_{jk}\right)}{\sum_{w=1}^{d_G} \exp\left(e_{iw}^{(1)} + \eta_{iw}\right) \sum_{m \in \mathcal{V}} \exp\left(e_{mw}^{(2)} + \xi_{mw}\right)} \mathbf{H}_j^{(l)}.
\end{aligned}
\tag{68}
$$

Let $\xi_{jk} = -\ln\left(\sum_{m \in \mathcal{V}} \exp(e_{mk}^{(2)})\right)$, i.e., $\exp\xi_{jk} = \left(\sum_{m \in \mathcal{V}} \exp(e_{mk}^{(2)})\right)^{-1}$, then

$$
\begin{aligned}
\left(\mathbf{A} \star_{\mathcal{G}} \mathbf{H}^{(l)}\right)[i] &= \sum_{j \in \mathcal{V}} \sum_{k=1}^{d_G} \frac{\exp\left(e_{ik}^{(1)} + \eta_{ik}\right) \exp\left(e_{jk}^{(2)}\right) / \sum_{m \in \mathcal{V}} \exp(e_{mk}^{(2)})}{\sum_{w=1}^{d_G} \exp\left(e_{iw}^{(1)} + \eta_{iw}\right) \sum_{m \in \mathcal{V}} \exp\left(e_{mw}^{(2)}\right) / \sum_{m \in \mathcal{V}} \exp(e_{mw}^{(2)})} \mathbf{H}_j^{(l)} \\
&= \sum_{j \in \mathcal{V}} \sum_{k=1}^{d_G} \frac{\exp\left(e_{ik}^{(1)} + \eta_{ik}\right)}{\sum_{w=1}^{d_G} \exp\left(e_{iw}^{(1)} + \eta_{iw}\right)} \frac{\exp\left(e_{jk}^{(2)}\right)}{\sum_{m \in \mathcal{V}} \exp(e_{mk}^{(2)})} \mathbf{H}_j^{(l)}
\end{aligned}
\tag{69}
$$

If $\boldsymbol{\eta}_i = \vec{\mathbf{0}}$, then the above Eq. 69 implies that,

$$
\mathbf{A} \star_{\mathcal{G}} \mathbf{H}^{(l)} = \text{Softmax}\left(\mathbf{E}_1\right) \text{Softmax}\left(\mathbf{E}_2^\top\right) \mathbf{H}^{(l)}.
\tag{70}
$$

This approach not only facilitates the calculation of similarity but also ensures the implementation of the multiplicative union law without activation after the inner product, thereby reducing the complexity to $\mathcal{O}\left(d_G^2 N\right)$ $(d_G \ll N)$ about linear complexity of nodes number $N$.

### A.3 Linear Combinations of Matrices to Raise Rank

**Theorem 2.** *The rank upper bound raising property of matrix addition.*

For any matrices $\mathbf{M}_1, \mathbf{M}_2, \ldots, \mathbf{M}_K \in \mathbb{R}^{D_1 \times D_2}$, their exists,

$$
\text{Rank}\left(\sum_{k=1}^{K} \mathbf{M}_k\right) \leq \sum_{k=1}^{K} \text{Rank}\left(\mathbf{M}_k\right).
\tag{71}
$$

***Proof.*** We can prove the theorem by means of chunked matrices. In fact,

$$
\begin{aligned}
\sum_{k=1}^{K} \text{Rank}\left(\mathbf{M}_k\right) &= \text{Rank}\left(\begin{bmatrix} \mathbf{M}_1 & & & \mathbf{0} \\ & \mathbf{M}_2 & & \\ & & \ddots & \\ \mathbf{0} & & & \mathbf{M}_k \end{bmatrix}\right) \\
&= \text{Rank}\left(\begin{bmatrix} \mathbf{M}_1 & \mathbf{M}_1 + \mathbf{M}_2 & \cdots & \sum_{k=1}^{K} \mathbf{M}_k \\ & \mathbf{M}_2 & & \sum_{k=2}^{K} \mathbf{M}_k \\ & & \ddots & \vdots \\ \mathbf{0} & & & \mathbf{M}_k \end{bmatrix}\right) \\
&\geq \text{Rank}\left(\sum_{k=1}^{K} \mathbf{M}_k\right).
\end{aligned}
\tag{72}
$$

$\square$

## B    Spatiotemporal Position Encoding

Spatiotemporal position encoding aims to distinguish data points by assigning them learnable embeddings that encode spatial and temporal positions, typically through concatenation [53]. Building on prior work [49, 54], we employ spatiotemporal embedding techniques to incorporate informative priors such as time-of-day and day-of-week. Integrating these meaningful representations into the model enhances its learning capability through effective positional prompting.

Concretely, we utilize three learnable positional encoding embeddings: spatial embedding $\mathbf{P}_S \in \mathbb{R}^{N \times d}$, timestep of day embedding $\mathbf{P}_T \in \mathbb{R}^d$, and day-of-week embedding $\mathbf{P}_D \in \mathbb{R}^d$. Spatial embedding $\mathbf{P}_S$ assigns a learnable embedding to each node to dynamically capture the spatial property of nodes. The timestep-of-day embedding $\mathbf{P}_T \in \mathbb{R}^d$ and day-of-week embedding $\mathbf{P}_D \in \mathbb{R}^d$ allocate corresponding learnable embeddings to each time step, dynamically extracting the periodicity of the spatiotemporal data. We use element-wise addition to form our spatiotemporal position encoding $\mathbf{P}$ as follows,

$$\mathbf{P} = \mathbf{P}_S + \mathbf{P}_T + \mathbf{P}_D \in \mathbb{R}^{N \times d}. \tag{73}$$

Unlike existing approaches that rely on concatenation, our addition-based method is theoretically equivalent to concatenation in terms of representation capacity, but offers greater computational efficiency. This approach serves two key purposes: (1) it avoids increasing the dimensionality of the intermediate hidden states, and (2) it potentially reduces the number of hyperparameters associated with the positional embeddings used in concatenation-based methods. The theoretical justification for this equivalence is provided in Appendix B.1.

### B.1    Equivalence Between Addition and Concatenating for Spatiotemporal Position Encoding

> **Theorem 3.** *Equivalence between + and* $||$ *for Spatiotemporal Position Encoding*
>
> Let $\tilde{\mathbf{X}}$ be the input data, $\mathbf{P}'_S \in \mathbb{R}^{N \times d_S}, \mathbf{P}'_T \in \mathbb{R}^{N \times d_T}$, and $\mathbf{P}'_D \in \mathbb{R}^{N \times d_D}$ are spatiotemporal position encoding for concatenation with weight parameters $\mathbf{W}'_0 \in \mathbb{R}^{(d+d_S+d_T+d_D) \times d}$ then their exits $\mathbf{P}_S, \mathbf{P}_T, \mathbf{P}_D \in \mathbb{R}^{N \times d}$ and $\mathbf{W}_0 \in \mathbb{R}^{d \times d}$, such that,
>
> $$[\tilde{\mathbf{X}}||\mathbf{P}'_S||\mathbf{P}'_T||\mathbf{P}'_D]\mathbf{W}'_0 = \tilde{\mathbf{X}}\mathbf{W}_0 + \mathbf{P}_S + \mathbf{P}_T + \mathbf{P}_D \in \mathbb{R}^{N \times d}. \tag{74}$$

*Proof.* In fact, the weight parameter $\mathbf{W}'_0$ can be viewed as

$$\mathbf{W}'_0 = [\mathbf{W}_0^\top || \mathbf{W}_S^\top || \mathbf{W}_T^\top || \mathbf{W}_D^\top]^\top \in \mathbb{R}^{(d+d_S+d_T+d_D) \times d}, \tag{75}$$

where $\mathbf{W}_0 \in \mathbb{R}^{d \times d}, \mathbf{W}_S^\top \in \mathbb{R}^{d_S \times d}, \mathbf{W}_T^\top \in \mathbb{R}^{d_T \times d}, \mathbf{W}_D^\top \in \mathbb{R}^{d_D \times d}$ are the composition of the first dimension of $\mathbf{W}'_0$. Then their exists,

$$\begin{aligned}[\tilde{\mathbf{X}}||\mathbf{P}'_S||\mathbf{P}'_T||\mathbf{P}'_D]\mathbf{W}'_0 &= [\tilde{\mathbf{X}}||\mathbf{P}'_S||\mathbf{P}'_T||\mathbf{P}'_D] \times [\mathbf{W}_0^\top || \mathbf{W}_S^\top || \mathbf{W}_T^\top || \mathbf{W}_D^\top]^\top \\ &= \tilde{\mathbf{X}}\mathbf{W}_0 + \mathbf{P}'_S\mathbf{W}_S + \mathbf{P}'_T\mathbf{W}_T + \mathbf{P}'_D\mathbf{W}_D.\end{aligned} \tag{76}$$

Then let $\mathbf{P}_S = \mathbf{P}'_S\mathbf{W}_S, \mathbf{P}_T = \mathbf{P}'_T\mathbf{W}_T, \mathbf{P}_D = \mathbf{P}'_D\mathbf{W}_D$, and we have,

$$[\tilde{\mathbf{X}}||\mathbf{P}'_S||\mathbf{P}'_T||\mathbf{P}'_D]\mathbf{W}'_0 = \tilde{\mathbf{X}}\mathbf{W}_0 + \mathbf{P}_S + \mathbf{P}_T + \mathbf{P}_D \in \mathbb{R}^{N \times d}. \tag{77}$$

$\square$

## C    Related Work

**Deep learning in time series analysis**. Deep learning has shaped a rich and diverse ecosystem for a wide range of time series tasks, such as forecasting [55, 56, 57, 58, 59], classification [60], imputation [61], and anomaly detection [62, 63, 64]. In recent years, neural architectures tailored for temporal data have advanced rapidly. Notably, MLP-based models [65, 66, 67] have emerged as highly efficient and scalable solutions, offering lightweight yet competitive performance. Meanwhile, Transformer-based methods [68, 69] lead in modeling power and predictive accuracy. Alongside architectural innovation, there has been growing interest in optimization strategies specifically adapted

to time series, aimed at improving training stability and robustness [70, 71, 72]. Furthermore, an increasing body of work focuses on downstream applications, seeking to align time series modeling with practical, domain-driven goals [73, 74].

Within this broader landscape, spatiotemporal forecasting represents a specialized branch of time series prediction. In contrast to generic forecasting tasks that model only temporal dynamics, spatiotemporal prediction focuses on short-term behavior shaped by structured spatial relationships, where nodes or sensors display strong, learnable interdependencies—such as in traffic flow, air quality monitoring, or weather systems [75, 76]. The core challenge in this area is to jointly model temporal evolution and spatial coupling in dynamic environments, a problem that continues to drive innovations in model architecture and computational efficiency.

**Load balanced optimization strategy**. Previous work such as DeepSeek [32] also follows a similar principle by applying the concept of balance to the design of large language models, demonstrating that balanced expert allocation can enhance model performance. However, their study focuses primarily on language modeling tasks, which differ substantially from our setting. In contrast, our work targets traffic forecasting—a distinct spatiotemporal prediction problem. Beyond adapting this idea, we make two key contributions. First, we provide a theoretical analysis of the balancing optimization strategy, including formal derivations that strengthen the interpretability and principled foundation of the approach. Second, rather than directly transferring the balancing concept, we ground it in matrix rank theory to formally justify its effectiveness in adaptive graph learning. This theoretical formulation not only clarifies why balancing improves performance but also enhances the overall interpretability and rigor of the balance-based paradigm in our context.

## D Experiments

### D.1 Dataset Description

The description of used spatiotemporal datasets are shown in Table 4.

**Traffic Domain. PeMS0X datasets (where X = 3, 4, 7, 8)** and **PeMS-Bay** are provided by the PeMS (Performance Measurement System) operated by the California Department of Transportation (Caltrans). These datasets with general-scale record traffic sensor data from multiple highway regions across California, with a sampling frequency of 5 minutes. The four larger-scale datasets **SD, GBA, GLA** and **CA** collectively referred to as LargeST, are also sourced from the PeMS system. The temporal resolution of these datasets is aggregated to 15 minutes in our experiments and we only choose the year 2019 in experimental comparison corresponding to current works [37], and they range in scale from 716 to 8,600 sensor nodes. **XTraffic** represents an even larger spatiotemporal system, comprising 16,972 nodes. Some of the above traffic dataset has capturing traffic flow, speed, and occupancy. More details are in Table 4.

**Energy. Electricity** dataset is a widely used benchmark for multivariate time series forecasting tasks. It records the hourly electricity consumption of 321 users or regions from 2012 to 2014, with a temporal granularity and sampling frequency of one hour. **UrbanEV** is a real-world dataset collected from 18,061 public charging stations in Shenzhen over a one-month period (from September 1, 2022 to August 31, 2023). The data is aggregated into 1,682 spatial regions. Temporally, the dataset has a time granularity of 5 minutes, resulting in a total of 8,640 time steps. Spatially, it covers 247 traffic analysis zones (nodes), forming a structured graph representation of urban electric vehicle charging demand.

**Meterology. Chinese Cities Air Quality** (CCAQ) dataset comprises AQI data and corresponding meteorological attributes from 209 cities in China mainland, spanning twenty-eight months (January 1, 2016, to April 30, 2019) with hourly temporal resolution. For our air quality forecasting model, we still focus on $PM_{2.5}$ as main prediction object. **KnowAir** dataset comprises $PM_{2.5}$ measurements and corresponding meteorological attributes from 184 cities in China mainland, spanning four years (January 1, 2015, to December 31, 2018) with three hour granularity.

**Mobility. Beijing Weibo dataset** contains blog check-in data received from 528 regions in Beijing through the Weibo application from January to December 2023. The Weibo application is a mainstream social media platform in China, with 590 million monthly active users as of 2024, offering extensive coverage. The data points are aggregated at one hour intervals. **Shanghai Mobile** dataset [42] comprises over 7.2 million call records generated by 9,481 mobile phones accessing the internet

Table 4: Description of the Spatiotemporal Datasets in the Experiments. M: Million ($10^6$). B: Billion ($10^9$). Data points is the multiplication of nodes and the total time steps.

| Domain | Datasets | # Nodes | # Edges | # Features | Time period | Frequency | Data Points |
|---|---|---|---|---|---|---|---|
| **Traffic** | PeMS03 | 358 | 546 | 3 | 09/01/2018 $\sim$ 11/30/2018 | 5 mins | 9.38 M |
| | PeMS04 | 307 | 338 | 5 | 01/01/2018 $\sim$ 02/28/2018 | 5 mins | 5.22 M |
| | PeMS07 | 883 | 865 | 3 | 05/01/2017 $\sim$ 08/06/2017 | 5 mins | 24.92 M |
| | PeMS08 | 170 | 276 | 5 | 07/01/2016 $\sim$ 08/31/2016 | 5 mins | 3.04 M |
| | PeMS-Bay | 325 | 2,369 | 3 | 01/01/2017 $\sim$ 06/30/2017 | 5 mins | 16.94 M |
| | LargeST-SD | 716 | 17,319 | 3 | 01/01/2017 $\sim$ 12/31/2021 | 5 mins | 0.38 B |
| | LargeST-GBA | 2,352 | 61,246 | 3 | 01/01/2017 $\sim$ 12/31/2021 | 5 mins | 1.24 B |
| | LargeST-GLA | 3,834 | 98,703 | 3 | 01/01/2017 $\sim$ 12/31/2021 | 5 mins | 2.02 B |
| | LargeST-CA | 8,600 | 201,363 | 3 | 01/01/2017 $\sim$ 12/31/2021 | 5 mins | 4.52 B |
| | XTraffic | 16,972 | 870,100 | 3 | 01/01/2023 $\sim$ 12/31/2023 | 5 mins | 1.78 B |
| **Energy** | Electricity | 321 | 101,323 | 5 | 01/01/2012 $\sim$ 12/31/2014 | 1 hours | 8.44 M |
| | UrbanEV | 1,682 | 1,989,840 | 5 | 09/01/2022 $\sim$ 08/31/2023 | 1 hours | 14.73 M |
| **Meterology** | KnowAir | 184 | 3,796 | 13 | 01/01/2015 $\sim$ 12/31/2018 | 3 hours | 2.15 M |
| | China City Air Qualtity | 209 | 4,321 | 10 | 01/01/2017 $\sim$ 04/29/2019 | 1 hours | 4.26 M |
| **Mobility** | Beijing Weibo | 528 | 244,942 | 3 | 01/01/2021 $\sim$ 01/01/2022 | 1 hours | 55.50 M |
| | Shanghai Mobile | 3,042 | 9,090,300 | 3 | 05/31/2014 $\sim$ 11/30/2014 | 1 hours | 13.36 M |
| | Milan Internet | 10,000 | 52,743,034 | 3 | 11/01/2013 $\sim$ 12/26/2013 | 1 hours | 43.93 M |

via 3,233 base stations from June 2014 to November 2014. The data time interval is also one hour. **Milan Internet** includes multiple mobile traffic features: outgoing calls (CALLOut), incoming calls (CALLIn), sent text messages (SMSOut), and received text messages (SMSIn). These features encompass mobility records collected over two months, from November 1, 2013, to January 1, 2014, across 400 regions. The data time interval is set to 1 hour. We use the Internet subdataset for fair performance comparison.

## D.2  Experiment Analysis

We compare the performance of MAGE and SOTA spatiotemporal baselines on common PeMS series datasets: PeMS0X (X=3,4,7,8) and PeMS-Bay. As shown in Table 5 and Table 6, MAGE basically dominates the optimal performance due to the powerful dynamic characterization capability of the multi-of-adaptive-graph module. MAGE is able to capture more accurate spatiotemporal dynamic pattern by dynamically selecting multiple efficient adaptive graph convolution results. However, the performance of adaptive graph convolutional methods, such as AGCRN, D$^2$STGNN, on small-scale datasets have been suboptimal since smaller datasets may lack sufficient spatiotemporal patterns to reliably train a single adaptive graph topology. In contrast, non-graph-convolutional spatial modeling approaches such as STNorm and STID have achieved impressive results, as their designs allow them to better capture temporal spatiotemporal patterns on limited data. Similarly, STWave with graph wavelet attention to learn the underlying graph structure has also demonstrated compelling performance on certain smaller datasets. Our proposed method, however, achieves universally superior predictive accuracy, outperforming almost every baselines. This improvement can be attributed to the introduction of a novel mixture-of-adaptive-graph-expert module. This module enables data-driven discovery of diverse underlying spatiotemporal graph topologies, thereby facilitating more precise spatiotemporal modeling.

Furthermore, we also report the performance comparison of MAGE on large-scale mobile datasets Shanghai Mobile with the 3,042 nodes and Milan Internet with 10,000 nodes. As shown in Table 6, thanks to the linear-complexity yet highly expressive Mixture-of-Adaptive-Graph-Experts (MAGE) structure, our approach maintains a clear lead over all competitors on these large-scale mobility benchmarks. However, quadratic-complexity adaptive graph convolution methods,such as AGCRN, GWNet, D$^2$STGNN, and Transformer-based graph learning models, such as STAEformer, can not be deployed on datasets of this scale due to their limited scalability. Even the linear-complexity GNN model struggles to match the performance of the classic MLP-based approach RPMixer, owing to its inherent low-rank limitations. In stark contrast, the efficient and high-performing MAGE module within our framework is able to deftly capture diverse and meaningful spatiotemporal latent graph structures, even on these massive datasets. This breakthrough in scalable spatiotemporal modeling is a testament to the elegance and power of our proposed approach.

Table 5: Performance comparisons on PeMS series datasets. The **best** and second best mean performance are in corresponding colors. The '/' marker indicates baseline is not applicable to this dataset due to the absence of key metadata (e.g., latitude and longitude). All experimental results are the average of five independent runs.

| Method | PeMS03 MAE | PeMS03 RMSE | PeMS03 MAPE | PeMS04 MAE | PeMS04 RMSE | PeMS04 MAPE | PeMS07 MAE | PeMS07 RMSE | PeMS07 MAPE | PeMS08 MAE | PeMS08 RMSE | PeMS08 MAPE | PeMS-Bay MAE | PeMS-Bay RMSE | PeMS-Bay MAPE |
|---|---|---|---|---|---|---|---|---|---|---|---|---|---|---|---|
| STGCN | 17.04 | 29.62 | 17.36 | 19.27 | 30.83 | 13.16 | 21.89 | 35.64 | 9.45 | 15.72 | 24.93 | 10.64 | 1.74 | 3.76 | 4.06 |
| DGCRN | 15.09 | 26.02 | 16.05 | 18.68 | 30.21 | 13.04 | 20.24 | 33.08 | 8.71 | 14.34 | 23.53 | 9.48 | 1.64 | 3.67 | 3.66 |
| AGCRN | 15.60 | 26.88 | 15.25 | 19.25 | 31.10 | 13.00 | 20.40 | 34.24 | 8.62 | 15.54 | 24.77 | 10.15 | 1.65 | 3.68 | 3.75 |
| GWNet | 14.76 | 25.35 | 15.38 | 18.81 | 30.29 | 13.06 | 19.92 | 32.84 | 8.62 | 14.20 | 23.13 | 9.53 | 1.61 | 3.61 | 3.63 |
| MTGNN | 15.31 | 25.95 | 15.04 | 19.20 | 31.81 | 13.26 | 20.97 | 34.20 | 8.90 | 15.22 | 24.09 | 9.89 | 1.63 | 3.66 | 3.62 |
| STNorm | 15.82 | 26.48 | 15.08 | 19.44 | 31.24 | 13.42 | 21.23 | 34.54 | 8.96 | 15.94 | 25.05 | 10.01 | 1.65 | 3.66 | 3.75 |
| STID | 15.37 | 26.39 | 16.57 | 18.29 | **29.74** | **12.45** | 19.54 | 32.54 | 8.28 | 14.19 | 23.28 | 9.25 | 1.70 | 3.86 | 3.91 |
| RPMixer | 16.19 | 25.91 | 15.96 | 21.11 | 33.56 | 14.88 | 23.95 | 38.77 | 10.63 | 17.33 | 27.47 | 11.31 | 1.91 | 4.36 | 4.27 |
| BigST | 15.30 | 25.77 | 16.54 | 18.42 | 29.96 | 12.92 | 20.31 | 33.57 | 8.57 | 14.19 | 23.26 | 9.29 | 1.65 | 3.58 | 3.77 |
| GSNet | 15.41 | 25.30 | 15.29 | 19.00 | 30.35 | 13.17 | 20.71 | 33.80 | 8.72 | 15.10 | 23.99 | 9.65 | 1.66 | 3.58 | 3.82 |
| STWave | 14.89 | 26.89 | 15.15 | 18.69 | 30.50 | 12.67 | 20.11 | 33.47 | 8.40 | 13.74 | 23.45 | **8.99** | 1.65 | 3.70 | 3.74 |
| STAEformer | 15.27 | 26.76 | 15.88 | 18.78 | 30.30 | 13.06 | 20.09 | 33.36 | 8.41 | 14.17 | 23.38 | 9.18 | 1.65 | 3.61 | 3.75 |
| D²STGNN | 14.84 | 25.41 | 15.17 | 18.61 | 30.13 | 12.82 | 20.33 | 33.23 | 8.73 | 14.36 | 23.46 | 9.32 | 1.62 | 3.69 | 3.68 |
| PatchSTG | / | / | / | / | / | / | / | / | / | / | / | / | 1.62 | 3.65 | 3.67 |
| **Ours** | **14.72** | **23.73** | **14.87** | **18.16** | 30.16 | 12.64 | **19.49** | **32.50** | **8.25** | **13.66** | **23.04** | 9.09 | **1.59** | **3.55** | **3.60** |

Table 6: Performance comparisons on large-scale mobile traffic datasets. The **best** and second best mean performance are in corresponding colors. The '-' marker indicates baseline incur out-of-memory issues even on minimum batch size. All experimental results are the average of five independent runs.

| Method | Shanghai Mobile MAE | Shanghai Mobile RMSE | Shanghai Mobile MAPE | Milan-Internet MAE | Milan-Internet RMSE | Milan-Internet MAPE |
|---|---|---|---|---|---|---|
| STGCN | 0.9607 | 1.7179 | 41.52 | 79.31 | 278.22 | 133.16 |
| DGCRN | - | - | - | - | - | - |
| AGCRN | - | - | - | - | - | - |
| GWNet | 0.9495 | 1.7337 | 39.32 | 46.40 | 159.78 | 58.56 |
| MTGNN | 0.9494 | 1.7131 | 40.72 | 66.97 | 230.80 | 117.67 |
| STNorm | 0.9735 | 1.7435 | 42.77 | 91.27 | 286.29 | 133.40 |
| STID | 0.9528 | 1.7094 | 40.62 | 47.24 | 152.97 | 50.29 |
| RPMixer | 1.0982 | 1.7852 | 53.25 | 44.90 | 140.79 | 55.74 |
| BigST | 0.9528 | 1.7079 | 41.38 | 46.44 | 143.73 | 63.60 |
| GSNet | 0.9541 | 1.7099 | 41.98 | 57.19 | 174.72 | 94.59 |
| STWave | - | - | - | - | - | - |
| STAEformer | - | - | - | - | - | - |
| D²STGNN | - | - | - | - | - | - |
| PatchSTG | 0.9646 | 1.7265 | 40.32 | 54.02 | 180.52 | 59.40 |
| **Ours** | **0.9356** | **1.6832** | **38.44** | **43.08** | **123.90** | **42.59** |

## D.3 Ablation Study

### D.3.1 Ablation Study on PeMS Series Datasets and Large-scale Mobility Datasets

We design the same following variants of our model to validate the soundness of the main component of our model on PeMS Series Datasets and Large-scale Mobility Datasets: '**w/o PE**' removes all the spatiotemporal position encoding embedding; '**w/o SE**' uses only feedforward networks as model backbone without spatial encoder; '**w/o Multi** ' leverages only one adaptive graph expert with $K = 1$; '**w/o LB**' reduces the load balanced optimization strategy in MAGE; '**w/o Sparse**' sums up all output of alternative graph convolution. The ablation study results presented in Figure 4 provide compelling insights. The performance degradation observed in both the w/o LB' and w/o Sparse' variants clearly indicates that sparse and balanced graph convolution operations outperform their dense counterparts and those without balancing, respectively. Furthermore, the 'w/o PE' variant also suffers from higher forecasting errors. This underscores the importance of the learnable spatiotemporal position encoding, which enables the model to extract valuable and generalizable knowledge during the training process. Interestingly, the 'w/o SE' variant achieves the worst performance among all. This finding highlights the crucial role played by our mixture-of-adaptive graph convolution module in guiding the model to effectively recognize the dynamic spatiotemporal dependencies between nodes.

### D.3.2 Ablation Study on Spatiotemporal Position Encoding

In this section, we conduct additional ablation study on spatiotemporal position encoding. Concretely, we construct multiple variants of MAGE in combinatorial ablation experiments by utilizing different combinations of spatiotemporal position embedding: 'w $\mathbf{P}_S$' is only with spatial position embedding. 'w $\mathbf{P}_T$' is only with timestep-of-day position embedding. 'w $\mathbf{P}_D$' is only with day-of-week position

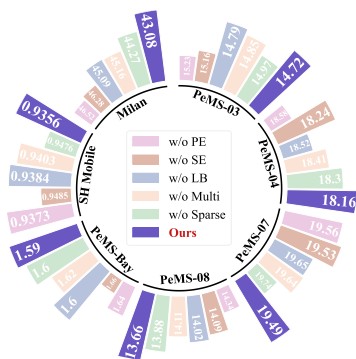

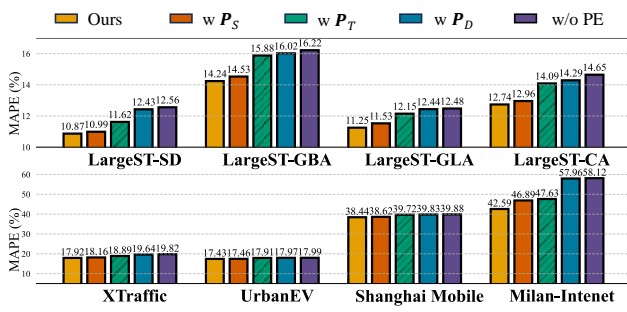

**(a) Ablation Study on PeMS series dataets and large-scale mobility datasets.**

**(b) Ablation study on different spatiotemporal position encodings.**

Figure 4: (a) Ablation Study on PeMS series dataets and large-scale mobility datasets. (b) Ablation study on different spatiotemporal position encodings.

embedding. As shown in Fig 4 (b), the addition of the spatial position embedding 'w $\mathbf{P}_S$' has indeed led to significant performance gains, as it enables the model to better emphasize and model the spatial positioning information. Both types of temporal position embeddings 'w $\mathbf{P}_T$' and 'w $\mathbf{P}_D$' have also proven valuable in helping the model capture temporal patterns more effectively. Notably, the 'w/o PE' variant, which lacks the spatiotemporal position encoding, exhibits the worst performance. This finding underscores the indispensable nature of the spatiotemporal position encoding in our framework. By seamlessly integrating the spatial and temporal position cues, the model is able to develop a more comprehensive understanding of the underlying data structures and dynamics.

### D.4 Experimental Evaluation of the Negative Role of ReLU in Adaptive Graph Learning

In our study, we theoretically prove that ReLU operation introduces additional edge noise in adaptive graph convolution in Appendix A.1. In this section, we empirically evaluate the negative effectiveness of ReLU in adaptive graph convolution. Concretely, we construct variants 'w/o ReLU' that reduces the ReLU operation before softmax normalization in the construction of adaptive graph of three classic STGNNs baselines: AGCRN [9], GWNet [7] and D²STGNN [23]. We conduct performance comparison experiments on deployable datasets in four domains: LargeST-SD, Electricity, KnowAir and Beijing Weibo. We still report the average results on five experiments. As shown in Table 7, the 'w/o ReLU' variants that reduces ReLU operation in adaptive graph construction gain better performances in all the datasets due to less edge noise generating proved in the Theorem 1 in Appendix A.1, possessing a relative improvement of at most 8.37%. Intriguingly, we have further

Table 7: Performance experiments on evaluating the negativity of ReLU in the adaptive graph convolution. We report the average results in five experiments. ↓ indicates the relative percentage decreasing regarding each methods itself.

| Methods | LargeST-SD | | | Electricity | | | KnowAir | | | Beijing Weibo | | |
|---|---|---|---|---|---|---|---|---|---|---|---|---|
| | MAE | RMSE | MAPE | MAE | RMSE | MAPE | MAE | RMSE | MAPE | MAE | RMSE | MAPE |
| AGCRN | 18.39 | 33.63 | 13.78 | 211.5 | 1847 | 16.95 | 16.34 | 24.81 | 63.26 | 0.8505 | 1.6998 | 33.68 |
| w/o ReLU | 18.29↓0.54% | 33.18↓1.34% | 13.32↓3.34% | 210.0↓0.71% | 1841↓0.32% | 15.53↓8.37% | 16.05↓1.77% | 24.36↓1.81% | 61.09↓3.43% | 0.8481↓0.28% | 1.6972↓0.15% | 33.40↓0.83% |
| GWNet | 18.07 | 29.97 | 12.70 | 200.3 | 1820 | 13.48 | 15.49 | 23.85 | 56.73 | 0.8315 | 1.6777 | 31.74 |
| w/o ReLU | 17.97↓0.55% | 29.33↓2.14% | 12.21↓3.86% | 199.0↓0.65% | 1755↓3.57% | 13.23↓1.85% | 15.49↓0.00% | 23.75↓0.42% | 56.63↓0.18% | 0.8292↓0.28% | 1.6665↓0.67% | 30.88↓2.71% |
| D²STGNN | 17.13 | 28.60 | 12.15 | 224.8 | 2110 | 17.46 | 15.39 | 24.31 | 55.41 | 0.8489 | 1.7216 | 31.89 |
| w/o ReLU | 16.99↓0.82% | 28.46↓0.49% | 12.03↓0.99% | 212.6↓5.43% | 2016↓4.45% | 17.33↓0.74% | 15.28↓0.71% | 24.16↓0.62% | 53.24↓3.92% | 0.8346↓1.68% | 1.7208↓0.05% | 31.35↓1.69% |

investigated the impact of incorporating ReLU activations within the adaptive graph model on the above datasets. As shown in Table 8, the results reveal that the introduction of ReLU has a detrimental effect on the training process, significantly increasing the average number of epochs required to converge. This observation aligns with our theoretical understanding that ReLU introduces undesirable edge noise into the adaptive graph generation process. This noise inherently impairs the model's ability to capture the true spatiotemporal dynamics, thereby compromising its generalization performance. Not only does the ReLU-induced edge noise limit the upper bound of the model's

achievable accuracy, but it also adversely impacts the training convergence speed, in some cases slowing it down by more than a factor of two. This is a crucial finding, as training efficiency is paramount for larger-scale real-world applications. These insights underscore the importance

Table 8: Average convergence epochs on evaluating the negativity of ReLU in the adaptive graph convolution in five experiments. The maximum allowable epochs are 300. ↓ indicates the relative percentage decreasing regarding each methods.

| Datasets | AGCRN | w/o ReLU | GWNet | w/o ReLU | D$^2$STGNN | w/o ReLU |
|---|---|---|---|---|---|---|
| LargeST-SD | 216 | 137$_{↓57.66\%}$ | 215 | 201$_{↓6.96\%}$ | 207 | 186$_{↓11.29\%}$ |
| Electricity | 300 | 287$_{↓4.53\%}$ | 208 | 185$_{↓12.43\%}$ | 56 | 52$_{↓7.69\%}$ |
| KnowAir | 41 | 39$_{↓5.13\%}$ | 34 | 34$_{↓0.00\%}$ | 36 | 33$_{↓9.10\%}$ |
| Beijing Weibo | 78 | 75$_{↓0.40\%}$ | 156 | 73$_{↓113.70\%}$ | 47 | 43$_{↓9.30\%}$ |

of principled architectural design choices when developing advanced spatiotemporal modeling frameworks. By carefully avoiding such pitfalls, our proposed MAGE approach is able to maintain its remarkable performance and training stability, even on the most challenging large-scale mobility datasets. Based on this, we have shown both theoretically and empirically that ReLU can have side effects on graph learning.

## D.5 Intrinsic Preference Study of Linear and Full-rank Adaptive Graph

Building on the Pareto-front analysis in Section 5.8, we observed that a pure linear-adaptive graph already outperforms—and is markedly faster than—any mixture that includes even a small fraction of full-rank adaptive graphs. This raises a natural follow-up question: if the model were free to decide, which family would it actually prefer? To answer it, we replace the global load-balancing constraint with an inner-balance mechanism that enforces equal activation only within each family (linear vs. full-rank) while letting the router autonomically allocate total capacity between the two under a equal 8:8 configuration. As shown in Fig. 5, the model self-assigns 94.7% of its routing mass to linear experts and only 5.3% to full-rank ones, yielding both higher accuracy and better generalization. This behavior suggests that full-rank adaptive graphs are prone to overfitting and introduce redundant complexity, further validating the efficiency and representational sufficiency of the linear expert formulation adopted in MAGE.

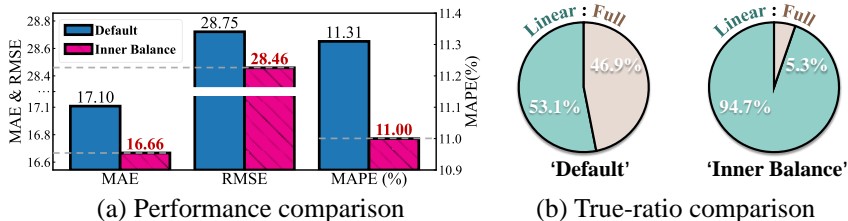

(a) Performance comparison        (b) True-ratio comparison

Figure 5: The comparison between 'default' load-balancing approach in MAGE and 'Inner balance' approach on the equal ratio setting 8:8.

## D.6 You Only Convolve Once

In this section, we demonstrate that the proposed MAGE achieves optimal performance with only a single graph convolution step. To this end, we construct variants of our model where each MAGE module is equipped with multi-step graph convolutions ranging from 1 to 10 layers. The results, shown in Figure 6, indicate that additional convolutional layers introduce only marginal computational overhead without yielding any significant performance gains. This observation confirms the strong expressive power of the learned multi-expert adaptive graph topology in capturing complex spatiotemporal dependencies.

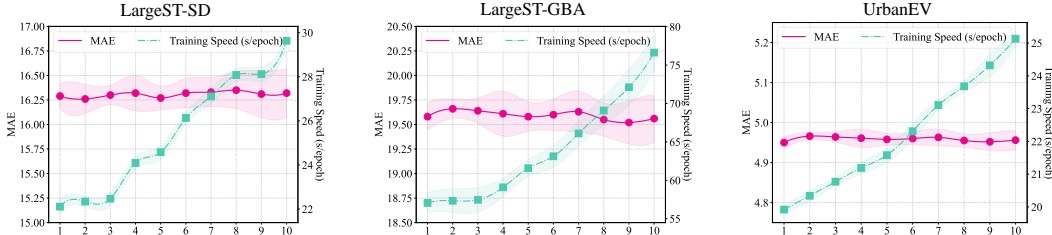

Figure 6: Performance and efficiency comparison of different layers of adaptive graph convolution in MAGE.

# E    Discussion and Future Work

In this section, we discuss the limitations of the current work and outline potential directions for future research. First, our model currently uses only a single-layer MLP to model temporal dependencies. In future work, we plan to explore more sophisticated temporal architectures, such as Transformers, to better capture complex dynamic patterns in the data. Second, existing spatiotemporal datasets often lack essential auxiliary information—such as community labels or ground-truth inter-regional connectivity—that is necessary for evaluating properties like connectivity. As a result, we are unable to thoroughly assess the effectiveness of the proposed adaptive graph in terms of structural consistency, expressiveness, and interpretability. Following Reviewer tEm8's suggestion, we will therefore focus on collecting datasets enriched with broader contextual information to further evaluate the model's strengths in aspects such as symmetry, normalization, and spectral properties. Finally, following Reviewer tEm8's suggestion, we also intend to extend the proposed method to general graph learning tasks [77, 78], including node classification, link prediction, and graph classification, to further validate its broad applicability.

