# OpenReview forum: "Less but More: Linear Adaptive Graph Learning Empowering Spatiotemporal Forecasting"
_NeurIPS.cc/2025/Conference — NeurIPS 2025 poster_

### Official Review · Reviewer_41Uo · 2025-06-16

**Clarity:** 3
**Significance:** 3
**Originality:** 3
**Rating:** 4
**Confidence:** 4

**Summary:**

This paper studies the problem of GNN-based spatiotemporal forecasting. Motivated by the deficiency of RELU in modeling dynamical graph topology used behind GNNs, this paper removes RELU and proposes a sparse yet balanced MOE strategy for efficiently approximating the full-rank Linear Adaptive Graphs. Extensive experiments on a large number of benchmark datasets show the superior performance of the proposed method against a sizeable collection of baselines.

**Questions:**

Please see the weaknesses.

**Ethical Concerns:**

["NO or VERY MINOR ethics concerns only"]

**Final Justification:**

After the rebuttal and discussion, I kept my positive score.

**Limitations:**

yes

**Quality:**

3

**Strengths And Weaknesses:**

Strengths:
- This paper is well motivated by illustrating the deficiency of RELU in modeling dynamical graph topology used behind GNNs.
- The proposed method is technically sound.
- The experiments compare the proposed method against a large collection of baselines on a number of benchmark datasets.

Weaknesses:
- Some design choices are not well justified, particularly the necessity of the kernel-inspired approximation. What if you directly apply MOE on the full-rank linear adaptive graphs, which completely avoid the low-rank bottleneck? Theoretically, this will be more expressive than your proposed method at the cost of higher computational cost. This trade-off should be rigorously discussed and studied in the experiments. What is the Pareto frontier of this trade-off?
- The proposed MAGE can be regarded as a base building block, which could be integrated with other spatiotemporal GNN models. Its advantages should be better justified by integrating with several other SOTA models. It would be interesting to see whether the performance improvement persists in these cases.

---

> ### Author Rebuttal · Authors · 2025-07-31
>
> Thank you very much for your kind recognition. Your time and effort have been invaluable in significantly improving the quality of our paper.
>
> ---
>
> ## **W1.**
>
> > a. The necessity of the kernel-inspired approximation
>
> Our motivation is to design a linear yet expressive adaptive graph structure, and these two goals are achieved through the two components of MAGE (MOE+kernel linear approximation). Specifically, **linearity is achieved via a kernel-inspired approximation method**, while the subsequent customized sparse and balanced MOE strategy helps restore the model’s rank to its upper bound (full rank), thus enhancing its representational power.
>
>
> > b. Only MAGE simultaneously satisfies both linear and full-rank properties in existing adaptive graphs.
>
> In fact, existing adaptive graph either possess full-rank characteristics (e.g., naive dot-product-based adaptive graph methods) but suffer from quadratic complexity, or emphasize linear complexity (e.g., relying solely on kernel-inspired dot-product approximations) yet are limited by low-rank constraints. Thus, **there are another full-rank linear adaptive graph case.** However, your question leads to another interesting one: what would happen if MOE were applied to a ~~linear~~ full-rank adaptive graph?
>
> > c. What if you directly apply MOE on the ~~linear~~ full-rank adaptive graphs, which completely avoid the low-rank bottleneck?
>
> To address this problem, we designed a MAGE variant called “MOE+ Nai”, where we combine a sparse and balanced MoE strategy with a naive (full-rank) adaptive graph. The hyperparameter for the number of experts is kept the same. We report its performance and efficiency on the SD dataset.
>
>
> | SD | MAE  | RMSE | MAPE | Memory  | Training Speed |
> |---------------|------|-------|------|----------|---------------|
> | MAGE (MOE+Linear kernel approximation)| **16.29** | **28.04** | **10.87** | **3,662 MB**| **22 s/epoch** |
> | MOE+Nai | 16.52 | 28.35 | 10.91 |4,308 MB| 46 s/epoch|
>
> Due to the quadratic complexity of the naive adaptive graph, this variant failed to improve performance and instead increased memory usage and training time. This is because our sparse and balanced MoE strategy is specifically designed for linear kernel adaptive graphs: by combining expert models that can capture different graph patterns, it enhances representational capacity and effectively achieves a full-rank approximation. However, when MoE is applied to a full-rank graph (which has already reached the upper bound of rank), it introduces redundant information, thus degrading learning performance. We provide further explanation in Section d(2).
>
>
>
> > d. This trade-off should be rigorously discussed and studied in the experiments. What is the Pareto frontier of this trade-off?
>
> (1). To address your question, we designed a set of experiments with 16 fixed graph experts, where each expert corresponds to an adaptive graph. According to the paper, 16 experts can achieve optimal performance. We then varied the mixing ratio of linear kernel adaptive graphs and naive adaptive graphs (full-rank), maintaining the original balanced and sparse expert activation constraints. The experimental results are as follows:
>
> | Linear : Full | MAE  | RMSE | MAPE | Memory  | Training Speed |
> |---------------|------|-------|------|----------|---------------|
> | 16:0 (Ours)          | **16.29** | **28.04** | **10.87** | **3,662 MB**   | **22 s/epoch**          |
> | 12:4          | 16.29 | 28.22 | 10.88 | 3,696 MB   | 24 s/epoch          |
> | 8:8           | 17.10 | 28.75 | 11.31 | 3,860 MB   | 30 s/epoch          |
> | 4:12          | 16.53 | 28.29 | 11.01 | 3,998 MB   | 35 s/epoch          |
> | 0:16 (MOE+Nai)         | 16.52 | 28.35 | 10.91 | 4,308 MB   | 46 s/epoch          |
>
> We observed that gradually increasing the proportion of full-rank adaptive graphs does not provide additional performance benefits aside from reducing computational efficiency. Therefore, we believe that the original MAGE module design has achieved a Pareto optimal state.
>
>
> (2). During this process, we observed an interesting phenomenon regarding the model’s expert selection behavior—its preference for linear adaptive graph experts over naive gadaptive raph experts. We used an 8:8 balanced ratio as a study example. Initially, the balance strategy forced the model to equally activate all 16 experts. In the variant, we introduced the concept of 'Inner Balance', allowing the model to adaptively choose between the linear graph expert family and the full-rank graph expert family, while maintaining balance in the activation of individuals within each family.
>
> |   | MAE  | RMSE  | MAPE  | Turth Ratio (Linear kernel vs Full-rank)    |
> |:---:|:--:|:-:|:-:|:-:|
> | Default | 17.10 | 28.75 | 11.31 | 53.1% : 46.9%   |
> | Inner Balance     | **16.66** | **28.46**  | **11.00** | 94.7% : 5.3%   |
>
> The results indicate that the Inner Balance variant not only improved performance but also revealed the model’s strong preference for linear kernel graph expert (94.7% vs. 5.3%). The default global balance (53.1% vs. 46.9%) inadvertently forces the model to retain overfitting full-rank experts, which can harm generalization ability. This further provides evidence of the incompatibility between MOE and full-rank adaptive graphs.
>
> ---
>
> ## **W2.**
>
>
> That’s a very innovative question. Using the SD dataset as an example, we integrated MAGE into the following four advanced spatiotemporal learning models: AGCRN, GWNet, STGCN, and STID. We reported their average performance over 12 time steps.
>
>
> | Model| MAE| RMSE| MAPE|
> |-------------|--------|--------|--------|
> | STGCN       | 19.27  | 33.57  | 13.49  |
> | +MAGE       | **19.19**  | **32.89**  | **12.70**  |
>
> | Model| MAE| RMSE| MAPE|
> |-------------|--------|--------|--------|
> | GWNet       | 18.07  | 29.97  | 12.70  |
> | +MAGE       | **18.13**  | **29.43**  | **12.26**  |
>
> | Model| MAE| RMSE| MAPE|
> |-------------|--------|--------|--------|
> | STID        | 18.03  |30.85   |12.18|
> | +MAGE       | **17.83**  |**30.42**    |**12.07**|
>
> | Model| MAE| RMSE| MAPE|
> |-------------|--------|--------|--------|
> | AGCRN       | 18.39  | 33.63  | 13.78  |
> | +MAGE       | **18.00**  | **32.21**  | **13.34**  |
>
> We observe that MAGE is able to improve the performance of these models to varying degrees, which demonstrates the good applicability of our method.

---

> > ### Comment · Reviewer_41Uo · 2025-08-05
> >
> > Thanks for the rebuttal. I will keep my positive score.

---

> > > ### Author Response · Authors · 2025-08-08
> > >
> > > Dear Reviewer 41Uo,
> > >
> > > We sincerely thank you for your incisive and constructive feedback. Your generous investment of time and expertise has elevated the manuscript’s clarity and rigor. We are deeply grateful for your invaluable guidance.
> > >
> > > Sincerely,
> > >
> > > The Authors of Submission 425

---

### Official Review · Reviewer_hrJD · 2025-06-25

**Clarity:** 3
**Significance:** 3
**Originality:** 3
**Rating:** 5
**Confidence:** 5

**Summary:**

This paper tackles the issues of edge noise and high complexity in traditional adaptive graph convolution methods by introducing a sparse mixture-of-experts linear graph learning approach. Theoretical analysis is provided to demonstrate the method’s effectiveness. Extensive experiments and comparisons with baseline models validate the efficacy and efficiency of the proposed model.

**Questions:**

Q1: Is there a way to avoid multiple experts learning redundant knowledge?

**Ethical Concerns:**

["NO or VERY MINOR ethics concerns only"]

**Final Justification:**

The authors have adequately addressed my concerns, and I choose to maintain a positive evaluation.

**Limitations:**

yes.

**Paper Formatting Concerns:**

NA.

**Quality:**

4

**Strengths And Weaknesses:**

**Strengths:**

S1: This paper presents an novel integration of the mixture-of-experts mechanism with spatiotemporal graph learning, and designs some novel techniques with solid theoretical contributions.

S2: The experiments conducted on 15 datasets are informative and comprehensive. And a 20-fold increase in efficiency is impressive.

S3: The authors have open-sourced the code to facilitate reproducibility. Focusing on efficient model for large-scale data is a cutting-edge research direction.

**Weakness:**

W1: The numerical formatting in Table 2 is inconsistent. Some values use separators while others do not. There are also spelling errors. For example, in line 289, there is an extra space inside the parentheses: (60%). Figure 2(a) is suggested to be presented in tabular form to improve readability.

W2:  The paper argues that the RULE function may introduce noise, but further experimental validation is needed to support this claim.

W3: Some important spatiotemporal forecasting benchmarks should be included for further comparison, such as LLGformer [1].

W4: The existing mixture-of-experts models in the spatiotemporal data domain should be discussed [2][3].

[1] Jin, D., Huo, C., Shi, J., He, D., Wei, J., & Yu, P. S. (2025, April). Llgformer: Learnable long-range graph transformer for traffic flow prediction. In Proceedings of the ACM on Web Conference 2025 (pp. 2860-2871)

[2] Jiang W, Han J, Liu H, et al. Interpretable cascading mixture-of-experts for urban traffic congestion prediction. Proceedings of the 30th ACM SIGKDD Conference on Knowledge Discovery and Data Mining. 2024: 5206–5217.

[3] Li S, Cui Y, Zhao Y, et al. ST-MoE: Spatio-temporal mixture-of-experts for debiasing in traffic prediction. Proceedings of the 32nd ACM International Conference on Information and Knowledge Management. 2023: 1208–1217.

---

> ### Author Rebuttal · Authors · 2025-07-31
>
> > Thank you very much for your kind recognition. Your time and effort have been invaluable in significantly enhancing the quality of our paper. We truly appreciate your support and constructive feedback.
>
> ## **W1. Typos of numerical formatting**
>
> Thank you for this valuable suggestion. In the camera-ready version we will perform a meticulous proof-read of every numeric entry in the tables to ensure a uniform, consistent format throughout.
>
> ---
> ---
>
> ## **W2. Experiments of edge noise induced by ReLU**
>
> We conducted a comprehensive experimental study (reported in **Appendix D6: Experimental Evaluation of the Detrimental Effects of ReLU in Adaptive Graph Learning**), which confirms that removing the ReLU activation consistently improves both prediction accuracy and convergence speed. Specifically, we selected four representative models—GWNet, AGCRN, and D$^2$STGNN—that incorporate the adaptive graph convolution module, and evaluated them on four real-world datasets spanning diverse domains. The results provide strong empirical support for our theoretical claims. For your convenience, we reproduce a subset of these results below.
>
> |  |    MAE |    RMSE |    MAPE |
> |:-:|:-:|:-:|:-:|
> | AGCRN    | 18.39 | 33.63 |  13.78|
> | w/o ReLU   |**18.29** | **33.18**| **13.32**|
>
> |             |    MAE |    RMSE |    MAPE |
> |:-:|:-:|:-:|:-:|
> | GWNet    | 18.07  | 29.97 |  12.70  |
> | w/o ReLU   | **17.97**| **29.33**|**12.21**|
>
> |             |    MAE |    RMSE |    MAPE |
> |:-:|:-:|:-:|:-:|
> |D$^2$STGNN|17.13|28.60|12.15|
> | w/o ReLU   |**16.99**|**28.46**|**12.03**|
>
> ---
> ---
>
> ## **W3. More Spatiotemporal Baselines**
>
> In the main experiments, we have already covered **17** open-source and publicly available advanced spatiotemporal models. Unfortunately, the code for LLGformer [1] has not been released. To address your concern, we introduce STGformer [2] as an additional baseline and evaluate it on four datasets across three domains. The results are reported below.
>
> | Model      |        | SD      |         |        | CA      |         |             | KnowAir      |              |             | UrbanEV      |              |
> |:------------:|:--------:|:---------:|:---------:|:--------:|:---------:|:---------:|:-:|:--:|:--:|:--:|:---:|:--:|
> |            |    MAE |    RMSE |    MAPE |    MAE |    RMSE |    MAPE |         MAE |         RMSE |         MAPE |         MAE |         RMSE |         MAPE |
> | STGformer  | 18.90  | 31.65   | 12.76   | 21.90  | 35.29   | 17.01   | 15.91       | 24.43        | 58.11        | 5.13        | 11.09        | 18.51        |
> | MAGE       | **16.29**  | **28.04**   | **10.87**   | **17.37**  | **29.37**   | **12.47**   | **15.36**       | **23.42**        | **52.77**        | **4.95**        | **11.00**        | **17.43**        |
>
>
>
> ---
> ---
>
> ## **W4. More discussion about MoE in Spatiotemporal Learnings**
>
> - **CP-MoE** [3] targets the *spatiotemporal congestion classification* task. Its MoE design relies on two graph experts that separately learn periodic and non-periodic congestion patterns.
>
> - **ST-MoE** [4] tackles *learning imbalance* in spatiotemporal tasks by stacking multiple STGNNs into an MoE network, allowing each node to select an appropriate STGNN.
>
> In contrast, our MoE module aims to "**restore the rank upper bound of linear adaptive graph convolution**": we employ a minimal set of graph convolutions to mitigate the inherent low-rank bottleneck commonly found in linear-complexity designs. Additionally, **we introduce a sparse and balanced modulation strategy to regulate the learning behavior of individual experts**. As a result, **both the objective and mechanism of our approach are fundamentally distinct from those of the aforementioned works**. Unfortunately, no open-source implementations of CP-MoE or ST-MoE are currently available, making direct performance comparisons infeasible. We commit to including a detailed discussion of these models in the next version of the manuscript.
>
>
>
>
> ---
> ---
>
> ## **Q1. Avoiding Knowledge Redundant in Experts.**
>
> We achieve this goal by integrating a load balancing mechanism to ensure that each graph expert is utilized evenly during training, while sparse gating enables each node to select only the most suitable experts. Together, these mechanisms encourage each expert to focus on capturing distinct underlying patterns. Additionally, based on matrix theory, we carefully control the number of experts to further prevent knowledge redundancy.
>
> Ref:
>
> [1] Jin, D., Huo, C., Shi, J., He, D., Wei, J., & Yu, P. S. (2025, April). Llgformer: Learnable long-range graph transformer for traffic flow prediction. In Proceedings of the ACM on Web Conference 2025 (pp. 2860-2871)
>
> [2] Wang H, Chen J, Pan T, et al. Stgformer: Efficient spatiotemporal graph transformer for traffic forecasting[J]. arXiv preprint arXiv:2410.00385, 2024.
>
> [3] Jiang W, Han J, Liu H, et al. Interpretable cascading mixture-of-experts for urban traffic congestion prediction. Proceedings of the 30th ACM SIGKDD Conference on Knowledge Discovery and Data Mining. 2024: 5206–5217.
>
> [4] Li S, Cui Y, Zhao Y, et al. ST-MoE: Spatio-temporal mixture-of-experts for debiasing in traffic prediction. Proceedings of the 32nd ACM International Conference on Information and Knowledge Management. 2023: 1208–1217.

---

> > ### Comment · Reviewer_hrJD · 2025-08-04
> >
> > Thanks for your response, which addressed my concerns. I will maintain my positive score.

---

> > > ### Author Response · Authors · 2025-08-08
> > >
> > > Dear Reviewer hrJD,
> > >
> > > We sincerely thank you for the invaluable comments that sharpened our work and enriched our perspective. Your guidance is deeply appreciated.
> > >
> > > Sincerely,
> > >
> > > The Authors of Submission 425

---

### Official Review · Reviewer_r3iW · 2025-06-26

**Clarity:** 3
**Significance:** 3
**Originality:** 4
**Rating:** 5
**Confidence:** 5

**Summary:**

The authors introduce an efficient sparse yet balanced mixture-of-experts strategy to tackle the complexity associated with current adaptive graph learning methods. They underpin the validity of their approach with comprehensive theoretical analyses. Compared with 14 spatiotemporal models across 17 datasets, their model demonstrates remarkable performance, while achieving significant improvements in memory efficiency, with a 10× reduction, and training efficiency, with a 20× increase.

**Questions:**

What is the model's potential for scalability on extremely large-scale spatiotemporal graphs?

**Ethical Concerns:**

["NO or VERY MINOR ethics concerns only"]

**Final Justification:**

The authors' response has addressed my concerns.

**Limitations:**

Yes, the authors have discussed limitations.

**Quality:**

4

**Strengths And Weaknesses:**

Strengths:

S1. The proposed model is reasonably motivated and logically well-designed, effectively addressing various challenges step-by-step.

S2. The idea of sparsing and balancing among the mixture of experts is admirable.

S3. This paper presents new theoretical insights.

S4. Extensive experiments including 17 datasets and 14 baselines prove the competitive performance and efficiency of their model.


Weakness:

W1. The authors have not provided an analysis or explanation for why PatchSTG fails to run on certain datasets.

W2. Spatiotemporal forecasting is a subset of time series, necessitating further discussion around additional time series forecasting models, such as PatchTST and TimeMixer.

W3. Is it possible to achieve performance gains by expanding the number of experts?

Ref:

[1] Nie, Y., Nguyen, N. H., Sinthong, P., & Kalagnanam, J. (2022). A time series is worth 64 words: Long-term forecasting with transformers. arXiv preprint arXiv:2211.14730.

[2] Wang, S., Wu, H., Shi, X., Hu, T., Luo, H., Ma, L., ... & Zhou, J. (2024). Timemixer: Decomposable multiscale mixing for time series forecasting. arXiv preprint arXiv:2405.14616.

---

> ### Author Rebuttal · Authors · 2025-07-31
>
> > Thank you very much for your kind recognition. Your time and effort have been invaluable in significantly improving the quality of our paper. We truly appreciate your support and constructive feedback.
>
> ## **W1. PatchSTG issues**
>
> We apologize for any confusion caused. PatchSTG model requires the latitude and longitude information of each node as additional input to determine which patch each node belongs to. However, due to the potential absence of such geographical data in publicly available datasets, the input requirements of the PatchSTG model cannot be met. Consequently, it fails to execute on certain datasets.
>
> ---
> ---
>
> ## **W2. Time series baselines**
>
> Time series models perform poorly in spatiotemporal prediction tasks because they only model dependencies between time steps while ignoring the crucial spatial dependencies among nodes [1].
>
> To address your question, we compared the performance of the MAGE model with time series models based on different architectures, including the Transformer-based PatchTST, the convolution-based TimeMixer, and the MLP-based DLinear, on the SD and KnowAir datasets, as shown below.
>
> | Model       |        |      SD |         |             |      KnowAir |              |
> |-------------|--------|---------|---------|-------------|--------------|--------------|
> |             |    MAE |    RMSE |    MAPE |         MAE |         RMSE |         MAPE |
> | PatchTST    | 48.50  | 74.20   | 30.84   | 19.04       | 29.28        | 61.66        |
> | TimeMixer   | 40.68  | 59.89   | 35.22   | 16.78       | 25.85        | 65.38        |
> | DLinear     | 49.82  | 75.43   | 39.96   | 17.06       | 26.20        | 64.53        |
> | MAGE(Ours)  |**16.29**  | **28.04**  | **10.87**  | **15.36**      | **23.42**        |**52.77**       |
>
> The results demonstrate that MAGE still exhibits the best performance on the aforementioned datasets, validating the effectiveness of our model.
>
> [1] Shao Z, Wang F, Xu Y, et al. Exploring progress in multivariate time series forecasting: Comprehensive benchmarking and heterogeneity analysis[J]. IEEE Transactions on Knowledge and Data Engineering, 2024.
>
> ---
> ---
>
> ## **W3. Performance of expanding more experts.**
>
> Blindly increasing the total number of experts does not improve performance; on the contrary, it may lead to overfitting the training data and result in decreased generalization ability. According to our matrix theory, once the number of experts exceeds a certain threshold, the model’s expressive power reaches its upper limit. As shown in Figure 2(b) in Section 5.3 of this paper, we systematically varied both quantities and reported the corresponding results. The experimental findings indicate that when the number of experts is too large, model performance actually drops.
>
> ---
> ---
>
> ## **Q. Potential for scalability**
>
> MAGE exhibits a computational complexity that scales linearly with the number of nodes, demonstrating strong theoretical scalability. Furthermore, we evaluate MAGE on XTraffic—the largest publicly available spatiotemporal dataset to date, containing 16,972 nodes. The results show that MAGE still achieves the best accuracy, further validating its excellent scalability. In contrast, several more complex models fail to run due to their high computational demands. For convenience, we present below the performance comparison with several state-of-the-art models.
>
> |    |    MAE |    RMSE |    MAPE |
> |-------------|--------|---------|---------|
> | PatchSTG    |10.63|20.86| 19.41 |
> |STID|  11.62 | 22.41 | 19.84 |
> | RPMixer| 16.68| 43.64| 32.74|
> | MAGE(Ours)  | **10.24**  |**20.48** |**17.92**|

---

> > ### Comment · Reviewer_r3iW · 2025-08-04
> > **Thanks for the rebuttal**
> >
> > I have checked the authors' responses which address my concerns. Thus, I would like to maintain my rating as acceptance.

---

> > > ### Author Response · Authors · 2025-08-08
> > >
> > > Dear Reviewer r3iW,
> > >
> > > Thank you for your thorough review and constructive feedback, which have significantly improved our manuscript. We appreciate your expertise and time.
> > >
> > > Sincerely,
> > >
> > > The Authors of Submission 425

---

### Official Review · Reviewer_tEm8 · 2025-07-02

**Clarity:** 2
**Significance:** 2
**Originality:** 3
**Rating:** 2
**Confidence:** 4

**Summary:**

This paper introduces Mixture-of-Adaptive Graph Experts (MAGE), a framework for adaptive graph learning in spatiotemporal forecasting. The method uses a mixture-of-experts approach, where multiple learned graphs are combined using node-specific affinity weights. MAGE aims to reduce computational complexity and improve flexibility compared to traditional adaptive graph methods. The paper provides comprehensive theoretical analysis and experimental results on several real-world datasets to support its approach.

**Questions:**

For questions, please see the above raised Weakness 1 - Weakness 4.

Aside from W1-W4, I also have another question about the storyline of this paper.
This paper **has unclear motivation**. **The storyline in the Introduction is not convincing and lacks logical coherent, particularly in the introduction section.** The authors first claim that the commonly used ReLU activation function in adaptive graph learning disproportionately amplifies negative edge weights and reinforces noisy edges. To address this, they propose replacing ReLU with a kernel-based approximation scheme. However, they then argue that this approximation introduces a low-rank bottleneck in the learned adjacency matrix, which subsequently motivates the introduction of their Mixture-of-Experts (MAGE) framework. Thus, my question is: Was the motivation for introducing MAGE simply to counteract the side effects introduced by using a kernel-based approximation scheme as a replacement for the ReLU activation function?
This sequence of arguments is problematic: if the main issue with ReLU is noise amplification, it is unclear why the only possible alternative must be a kernel-based function that suffers from low-rank limitations. The authors do not sufficiently justify why other activation or similarity functions that avoid both noise amplification and low-rank issues could not be used instead. As a result, the motivation for MAGE appears ad hoc and unconvincing, undermining the overall contribution and logical flow of the paper.

**Ethical Concerns:**

["NO or VERY MINOR ethics concerns only"]

**Final Justification:**

The paper introduces MAGE for efficient adaptive graph learning in spatiotemporal forecasting, with strong empirical results. The rebuttal provided additional baselines, partial graph-level evaluation, and clarification that MAGE and the ReLU-related modification are independent contributions. However, the core concerns remain: the initial framing of prior work still overlooks recent efficient adaptive graph methods; global structural consistency and interpretability are not convincingly addressed; graph-level analysis is limited and largely indirect; and the motivation linking design choices remains unconvincing. Despite the added clarifications, these fundamental weaknesses persist, so my original reject rating is upheld.

**Limitations:**

Please see the above raised Weakness 1 - Weakness 4

**Paper Formatting Concerns:**

No concerns associated with paper formatting. The figures and tables look good. The formatting is elegant.

**Quality:**

2

**Strengths And Weaknesses:**

**Strengths:**

(1) The proposed MAGE framework addresses the computational bottleneck of traditional adaptive graph learning by introducing a kernel-based approximation, which reduces the graph construction complexity from quadratic to linear with respect to the number of nodes. This design is particularly practical for large-scale spatiotemporal datasets and is supported by both theoretical analysis and empirical benchmarks.

(2) By using a mixture-of-experts mechanism, MAGE allows for node-specific combinations of multiple learned graphs. This provides additional flexibility in capturing diverse and dynamic spatiotemporal dependencies, which could be beneficial in heterogeneous or complex forecasting scenarios.

(3) The paper conducts extensive experiments across 17 real-world datasets from multiple domains, and includes ablation studies and efficiency comparisons. These experiments demonstrate that the method can achieve competitive, and sometimes state-of-the-art, results in both accuracy and efficiency when compared to a wide range of existing baselines.


**Weaknesses:**

**W1:** A major weakness of the paper is its oversimplified and outdated characterization of existing adaptive graph learning methods. The authors claim that dynamic graphs are typically constructed via pairwise embedding inner products, resulting in $O(N^2)$ computational complexity. However, this only describes a subset of early adaptive graph models such as AGCRN and GWNet. In fact, many recent adaptive graph construction approaches Implicit learning-based methods, avoiding explicit dense adjacency computation altogether. Furthremore, a large number of modern methods already employed top-K selection, sparse routing, or on-the-fly edge generation, achieved much lower complexity in practice [1] [2]. Therefore, presenting the $O(N^2)$ embedding inner product approach as representative of the entire field is inaccurate and ignores the significant methodological diversity and existing research progress that have been made in adaptive graph learning.



**References:**


[1] Huang, Siyuan, et al. "Tailoring self-attention for graph via rooted subtrees." *Advances in Neural Information Processing Systems* 36 (2023): 73559-73581.

[2] Qian, Chendi, et al. "Probabilistic Graph Rewiring via Virtual Nodes." *The Thirty-eighth Annual Conference on Neural Information Processing Systems*.





**W2: Limited Global Structural Consistency, Expressiveness, and Interpretability.** The design of MAGE enables each node to construct its own adaptive adjacency structure by selectively combining expert-generated graphs using node-specific affinity weights. While this approach allows for high local flexibility, it fundamentally breaks the consistency of a unified global graph structure. Unlike traditional GNNs, where all nodes share the same adjacency matrix. As a result, important global properties such as overall connectivity, cycles, community structure, or long-range dependencies may not be explicitly modeled or preserved. This lack of structural coherence can be particularly problematic in tasks where global graph topology or prior knowledge plays a crucial role.
 Additionally, the "personalized" and fragmented nature of the learned adjacency makes it extremely difficult to interpret or visualize the resulting graph, limiting its practical utility for scientific analysis, knowledge discovery, or downstream applications that require graph explanations. The method also lacks theoretical justification regarding its ability to model or recover key global attributes (such as symmetry, normalization, or spectral properties) that are well-studied in the GNN literature. Without systematic empirical or theoretical analysis on these aspects, it is unclear whether the proposed approach can generalize or perform robustly on problems where global graph characteristics are critical.



**W3: Lack of Direct Graph-Level Validation and Analysis.** The experimental validation in this paper is largely indirect. The majority of the experimental section focuses on demonstrating that using MAGE leads to improved spatiotemporal forecasting performance on certain datasets, as measured by standard prediction error metrics. However, such improvements only provide indirect evidence of the effectiveness of the proposed graph learning method. The paper lacks more direct graph-level validation from the perspective of graph structure analysis. For example, there is no comparison of the sparsity or connectivity patterns of the graphs constructed by MAGE versus other methods, to show whether MAGE successfully preserves the most important edges. Additionally, the paper does not include any visualization or alignment of the learned graphs with the true underlying system structure at time $t$, to demonstrate that the edges constructed by MAGE meaningfully correspond to real, strong connections in the data. This omission makes it difficult to assess whether the proposed method actually learns more accurate or interpretable graph structures.



**W4:** **Insufficient Empirical Evidence for the Claimed Drawbacks of ReLU Activation.** The authors claim that the use of ReLU activation in existing adaptive graph learning methods amplifies edge-level noise, resulting in the creation of spurious or unreliable connections in the learned adjacency matrix. However, this assertion is only supported by a simple theoretical analysis, without any dedicated empirical validation. The paper does not provide ablation studies, visualizations, or quantitative experiments to directly demonstrate the negative impact of ReLU-induced noise or to substantiate the practical advantages of their proposed alternative. As a result, the significance of this issue remains unconvincing.

---

> ### Author Rebuttal · Authors · 2025-07-31
>
> We sincerely thank you for your professional advice and the time you have dedicated. We have gained valuable insights, and based on your feedback, we can significantly enhance the quality of our paper.
>
> ---
> ---
>
> ## **W1.**
>
> The methods in [1, 2] are computationally efficient variants of the Transformer model, characterized by high model complexity and a large number of parameters. In contrast, the dot-product adaptive graph module requires only two learnable embeddings for pairwise dot-product calculations, making it significantly more lightweight and simplified. **This approach also offers good ease of integration and has consequently been widely adopted. For these reasons, we have chosen the dot-product adaptive graph learning method as the foundation of our research.**
>
> To demonstrate this through experiments, we addressed a critical limitation: the works [1, 2] focus on graph representation tasks and lack the temporal modeling elements needed for spatiotemporal prediction. To ensure a fair comparison, we integrated the graph learning methods from MAGE and [1, 2] into mature spatiotemporal prediction models GWNet and AGCRN by replacing their adaptive graph modules. We then assessed their performance and complexity, under optimal performance settings, on the SD dataset. The results are summarized below.
>
> || MAE|RMSE|MAPE|Memory (MB)|Training Speed (s/epoch)|
> |:-:|:-:|:-:|:-:|:-:|:-:|
> |GWNet| 18.07| 29.97| 12.70|8,990|83.56|
> |+STA| 19.13| 31.39| 12.83|9,737|81.18|
> |+IPR-MPNNs| 26.22| 41.49| 17.33|9,302|78.77|
> |+MAGE|**17.87**|**29.43**|**12.26**|**6,994**|**67.52**|
>
> || MAE|RMSE|MAPE|Memory (MB)|Training Speed (s/epoch)|
> |:-:|:-:|:-:|:-:|:-:|:-:|
> |AGCRN| 18.39| 33.63|13.78|8,100|80.58|
> |+STA| 18.05| 33.08 |13.61|8,746|76.32|
> |+IPR-MPNNs| 20.45|34.51|14.41|8,443|65.94|
> |+MAGE|**18.00**|**32.21**|**13.34**|**6,810**|**59.12**|
>
> We discovered that integrating these Transformer variants into the model actually led to decreased performance, along with a significant increase in the number of parameters and memory usage.
>
> Based on your suggestions, we will integrate these discussions and related recent advancements into the manuscript in future revisions. Thank you again for your advice.
>
> ---
> ---
>
> ## **W2.**
>
> Thank you for your valuable insights into graph learning, which will inspire our future research.
>
> > Allow me to clarify first: our research is focused solely on the specific downstream task of spatiotemporal prediction, and **does not aim to develop a general-purpose GNN applicable to various downstream tasks.**
>
> In the field of spatiotemporal forecasting, **(1) Datasets often lack essential ancillary information necessary for evaluating properties such as connectivity, which can limit the comprehensive analysis.** **(2) The effectiveness of traditional GNNs that rely on a shared global graph structure is being questioned. Currently, adaptive graph learning methods dominate the field.** The main reasons are as follows:
>
> - **Poor-quality or even non-constructable graph topology**. Unlike general graph learning settings, graphs in most spatiotemporal prediction datasets are not naturally occurring but are constructed using secondary measures such as geographical proximity. **These distance-based graphs fail to capture other critical global properties**, such as connectivity, cyclicity, and community structure. In some cases, even global graphs derived from geographic information are unavailable due to privacy concerns.
>
> - **There is a lack of relevant prior data to validate the advantages of GNNs with shared global graphs**. Most datasets lack the necessary prior knowledge, such as community labels or actual inter-regional connectivity, which hinders the evaluation of the advantages of GNNs with unified global graph in terms of properties like symmetry, normalization, or spectral attributes. **These limitations further restrict the direct assessment of model interpretability.**
>
> > We can take a different perspective: **perhaps we can integrate the adaptive graph learning method MAGE into GNNs based on unified global graph**.
>
> In this way, it can retain the advantages of MAGE while also inheriting the strengths of traditional GNNs. To explore this possibility, we have incorporated MAGE into GWNet and STGCN by performing multi-graph convolution using both the adaptive adjacency matrix and their global graphs. The following results demonstrate that MAGE can improve the performance of GNNs based on unified global graph, which inspires our future work in designing hybrid graph convolutional networks.
>
> |SD dataset|MAE|RMSE|MAPE|
> |:-:|:-:|:-:|:-:|
> |STGCN|19.27|33.57|13.49|
> |+MAGE|19.19|32.89|12.70|
> |GWNet|18.07|29.97|12.70|
> |+MAGE|**17.87**|**29.43**|**12.26**|
>
> ---
> ---
>
> ## **W3.**
>
> The ultimate goal of spatiotemporal forecasting is to improve prediction accuracy. **Beyond the core adaptive graph learning component, our MAGE model consists solely of simple fully connected layers; therefore, its performance gains are primarily attributable to the proposed adaptive graph learning method**. This is extensively validated by the ablation studies in Figure 2 of the paper. These two evidences directly prove the effectiveness of our graph learning approach.
>
> Thank you for your suggestion to include additional graph evaluations. We appreciate your feedback and would like to provide the following response.
>
> - Experiment
>
> A well-structured graph topology should effectively reflect the similarity among node labels. To evaluate the quality of the learned graph topologies, we first integrate three advanced methods—MAGE, [1, 2], and the original adaptive graph learning—into GWNet, thereby isolating the evaluation from potential difference factors introduced by other model components. We then report the mean squared error (MSE) between the generated adjacency matrices and the similarity matrix derived from node labels, providing a quantitative measure of how well the learned structures capture label coherence.
>
> | SD dataset  | Ours| STA | IPR-MPNNs |Original
> |:-:|:-:|:-:|:-:|:-:|
> | Similarity MSE | **0.00135**| 0.198| 0.389|0.00838|
>
> The adjacency matrix learned by our model shows the smallest difference with respect to the node label affinity, which provides a solid foundation for accurate prediction by the model.
>
> -  Visualization
>
> Due to the NIPS rebuttal policy prohibiting the inclusion of any images, we promise to add more visualizations of generated graphs in the final version. These cases will demonstrate the association between the learned adjacency matrices and the patterns of data distributions.
>
> ---
> ---
>
> ## **W4.**
>
> > **We have done this experiment!** As emphasized in line 115 of the manuscript, we provide experimental evidence for this theory, with the results shown in Appendix D.6.
>
> Specifically, we demonstrate across four datasets and multiple models employing the adaptive graph learning module that removing the ReLU activation function improves prediction performance, as shown in **Table 6**. Furthermore, as illustrated in Table 7, removing ReLU also accelerates model convergence. For convenience, we reproduce a subset of results on the SD dataset below. These findings provide strong empirical evidence supporting the effectiveness of our design choice.
>
>
> |  |    MAE |    RMSE |    MAPE |
> |:-:|:-:|:-:|:-:|
> | AGCRN    | 18.39 | 33.63 |  13.78|
> | w/o ReLU   |**18.29** | **33.18**| **13.32**|
>
> |             |    MAE |    RMSE |    MAPE |
> |:-:|:-:|:-:|:-:|
> | GWNet    | 18.07  | 29.97 |  12.70  |
> | w/o ReLU   | **17.97**| **29.33**|**12.21**|
>
> |             |    MAE |    RMSE |    MAPE |
> |:-:|:-:|:-:|:-:|
> |D2STGNN|17.13|28.60|12.15|
> | w/o ReLU   |**16.99**|**28.46**|**12.03**|
>
> （2）We added the ReLU function to the proposed MAGE model and compared its performance with the original MAGE, as shown in the table below. This result also serves as valid experimental evidence.
>
> |  | MAE |  RMSE | MAPE |
> |:-:|:-:|:-:|:-:|
> |Ours|**16.29**|**28.04**|**10.87**|
> |+ ReLU   |16.81|28.90|11.05|
>
> ---
> ---
>
> ## **Q1. MAGE has nothing to do with ReLU.**
>
> > MAGE is **Not** simply to counteract the side effects introduced by using a kernel-based approximation scheme as a replacement for the ReLU activation function!**
>
> **As emphasized in the abstract, introduction, and methods sections, MAGE and the ReLU noise theory improve two distinct components including in adaptive graph learning framework**—namely, RuLE and the vector dot product—and represent independent improvements. **Therefore, MAGE is not designed to mitigate any side effects that may arise from replacing the ReLU activation function with a kernel-based approximation scheme.** Specifically,
>
> - RuLE Noise Theory: This theory reveals that RuLE introduces noise, leading to performance degradation, and thus we remove it. **As demonstrated in W4, this modification does not introduce any side effects**; on the contrary, it brings performance improvements.
>
> - MAGE: MAGE is designed to address the challenge of high computational complexity in dot product operations. First, we propose kernel-based functions as efficient alternatives to dot products. Then, we introduce a sparse yet balanced mixture-of-experts system to enhance expressive power.
>
> To further avoid confusion, we will revise the last paragraph of the introduction and the methods section to highlight the relationship between the two, namely that we have made independent improvements to the two components of dot-product-based adaptive graph learning.
>
> Ref:
>
> [1] Huang, Siyuan, et al. "Tailoring self-attention for graph via rooted subtrees." Advances in Neural Information Processing Systems 36 (2023): 73559-73581.
>
> [2] Qian, Chendi, et al. "Probabilistic Graph Rewiring via Virtual Nodes." The Thirty-eighth Annual Conference on Neural Information Processing Systems.

---

> ### Author Response · Authors · 2025-08-04
> **Humbly Seeking Further Discussion and Guidance**
>
> Dear Reviewer tEm8,
>
> Thank you very much for your thorough review. If you have any further questions, we would be happy to discuss them with you.
>
> For your convenience, we have summarized our responses as follows:
>
> -  **For W1. More comparative baselines.** We used experiments to demonstrate the superiority of our model compared to the two works you are concerned with.
>
> - **For W2. Non-natural spatiotemporal graphs contain limited information.** Unlike in graph learning tasks, the graph of most spatiotemporal prediction datasets is constructed based on the geographical locations of nodes (not naturally derived), and therefore contains limited symmetry, normalization, or spectral properties.
>
> - **For W2. Lack of spatiotemporal data for evaluating global properties.** The lack of prior data in spatiotemporal datasets prevents the evaluation of certain global property advantages of GNNs with shared global graphs, such as community structure or connectivity. And GNNs with shared global graphs are gradually being replaced in the field of spatiotemporal prediction.
>
> - **For W3. Graph-level evaluation.** We present an experiment that demonstrates a stronger correlation between our generated graph and the node prediction values.
>
> -  **For W4. Experimental evidence for ReLU theory.** We clarified that we have already supported our claims about ReLU with experimental evidence in the paper, please refer to **Table 6** in the submission.
>
>
> - **For Q1. MAGE and ReLU noisy theory are independent.** We clarify the independent roles of ReLU noise theory and MAGE, which are respectively designed to address the limitations of two components in traditional adaptive graph learning.
>
>
> Once again, thank you for your professional feedback. **Based on your suggestions, we will make dedicated efforts:**
>
> (1). Based on your suggestions, we commit to making significant revisions to the paper, including adding more visual examples and analysis of adaptive graphs for graph-level validation and further discussing related works in the field of graph learning.
>
> (2). Your comments have inspired us to collect a new dataset with more spatiotemporal prior data to better align the developments of the spatiotemporal learning domain with the graph learning domain.
>
> We sincerely recognize how valuable your time and attention are, and we are profoundly thankful for any additional insights or advice you may share.
>
> Thank you again for your kindness and support.
>
> Best regards,
>
> The Authors of Submission 425

---

### Decision · Program_Chairs · 2025-09-17

**Decision:**

Accept (poster)

**Comment:**

This paper proposes MAGE for efficient spatiotemporal forecasting. The paper explores the validity of their approach with comprehensive theoretical analyses and compares with multiple spatiotemporal models across different datasets, and displays remarkable performance. The theoretical analysis of this paper is rigorous and the experimental evaluations are sufficient to justify the statements.

I'm leaning toward accepting the paper.